# The divergent mitotic kinesin MKLP2 exhibits atypical structure and mechanochemistry

Joseph Atherton[1†], I-Mei Yu[2†], Alexander Cook[1], Joseph M Muretta[3], Agnel Joseph[1], Jennifer Major[4], Yannick Sourigues[2], Jeffrey Clause[2], Maya Topf[1], Steven S Rosenfeld[4], Anne Houdusse[2*], Carolyn A Moores[1*]

[1]Institute of Structural and Molecular Biology, Birkbeck College, London, United Kingdom; [2]Structural Motility, Institut Curie, Centre National de la Recherche Scientifique, Unité Mixte de Recherche, Paris, France; [3]Department of Biochemistry, Molecular Biology, and Biophysics, University of Minnesota, Minneapolis, United Sates; [4]Department of Cancer Biology, Lerner Research Institute, Cleveland Clinic, Cleveland, United States

*For correspondence:
Anne.Houdusse@curie.fr (AH);
c.moores@mail.cryst.bbk.ac.uk (CAM)

[†]These authors contributed equally to this work

Competing interests: The authors declare that no competing interests exist.

**Abstract** MKLP2, a kinesin-6, has critical roles during the metaphase-anaphase transition and cytokinesis. Its motor domain contains conserved nucleotide binding motifs, but is divergent in sequence (~35% identity) and size (~40% larger) compared to other kinesins. Using cryo-electron microscopy and biophysical assays, we have undertaken a mechanochemical dissection of the microtubule-bound MKLP2 motor domain during its ATPase cycle, and show that many facets of its mechanism are distinct from other kinesins. While the MKLP2 neck-linker is directed towards the microtubule plus-end in an ATP-like state, it does not fully dock along the motor domain. Furthermore, the footprint of the MKLP2 motor domain on the MT surface is altered compared to motile kinesins, and enhanced by kinesin-6-specific sequences. The conformation of the highly extended loop6 insertion characteristic of kinesin-6s is nucleotide-independent and does not contact the MT surface. Our results emphasize the role of family-specific insertions in modulating kinesin motor function.

## Introduction

The success of mitosis depends on the intricate timing, precise localisation and regulated interactions between multiple spindle components. MKLP (Kif20A), a kinesin-6 family motor, is involved at several key stages of cell division, and in particular is essential for cytokinesis in mammals (*Hill et al., 2000*). MKLP2 interacts with and is regulated by a number of mitotic kinases including Cdk1/cyclin B prior to anaphase onset (*Hümmer and Mayer, 2009*; *Kitagawa et al., 2014*). MKLP2 is required for localisation of Chromosome Passenger Complex (CPC) components, including Aurora B kinase, to the midzone at anaphase (*Gruneberg et al., 2004*), which is in turn required for Kif4A and PRC1 function (*Nunes Bastos et al., 2013*). MKLP2 also participates in myosin II recruitment for cleavage furrow ingression (*Kitagawa et al., 2013*) and interacts with the cell membrane to promote abscission (*Fung et al., 2017*). In addition, it is a substrate for polo-like kinase 1 (Plk1), and the co-association at the midzone of MKLP2 and Plk1 is also required for cytokinesis (*Neef et al., 2003*). In the light of these activities and because MKLP2 is found to be highly expressed in tumour tissue (*Gasnereau et al., 2012*), inhibition of MKLP2 by antimitotics has been explored, and small molecule inhibitors of MKLP2 have been shown to perturb mammalian cytokinesis (*Tcherniuk et al., 2010*; *Labrière et al., 2016*). While there has been significant focus on dissecting cell cycle dependent function and regulation of MKLP, little is known about the molecular mechanisms that drive this

**eLife digest** Cells constantly replicate to provide new cells for growing tissues, and to replace ageing or defective cells around the body. Each new cell needs a copy of the genetic material, and a cellular structure called the mitotic spindle makes sure that this material is shared correctly when a cell divides in two. The spindle is built from protein filaments called microtubules, and the protein filaments grow and shrink as the mitotic spindle carries out its role. Many of these changes in the spindle are driven by proteins called molecular motors, which break down energy-rich molecules of ATP to power them as they walk along the filaments.

Kinesins, for example, are molecular motors that can move along microtubules and there are over 40 different kinesins encoded in the human genome. More than half of the human kinesins are involved in cell division including one called MKLP2. Little is known about MKLP2 but some earlier findings had suggested that it would behave very differently compared to other kinesins.

Understanding how a kinesin motor works requires studying it in complex with its microtubule tracks. Atherton, Yu et al. have now used a technique called cryo-electron microscopy – which is uniquely suited to looking at large and complicated samples in three dimensions – to observe how the motor in MKLP2 changes shape as it works. This revealed that, while MKLP2 works in a fundamentally similar way to other kinesins, many aspects of its molecular mechanism are highly unusual. These include how it binds to the microtubule, how it interacts with ATP and how it generates force.

These findings show that there is much greater diversity in the molecular mechanisms of the kinesins involved in cell division than was previously thought. Several anticancer drugs target kinesins to stop cells dividing and so this diversity may make it easier to target only certain kinesins with drugs, which in turn would have fewer side effects. First, though, it will be important to find out how the unusual mechanism of MKLP2 coordinates and influences other components of the spindle to reveal a fuller picture of what happens when cells replicate.

function. MKLP2 shares only approximately 35% sequence identity with kinesin-1 (Kin1), which suggests that much of what is known about kinesin molecular function may not necessarily apply to this motor (*Hizlan et al., 2006*).

Little is known about the molecular and mechanistic properties of MKLP2. A related kinesin family member, the kinesin-6 MKLP1 - part of the centralspindlin complex that is also essential for cytokinesis - is a plus-end directed motor (*Nislow et al., 1992*), as is Kif20B, another kinesin-6 (*Abaza et al., 2003*). The centralspindlin holocomplex has anti-parallel microtubule (MT) bundling capability (*Davies et al., 2015*) consistent with its localisation and function in the spindle midzone. Functional diversification within the kinesin superfamily is derived in part from variation within kinesin motor domain itself. Kinesin-6s are particularly intriguing in this context because they are up to ~40% larger than the canonical Kin1 motor domain (*Figure 1A*) due to a number of unique loop insertions. Within the mammalian MKLP2 motor domain sequence, these unique features include a ~ 60 residue N-terminal extension, insertions in loop 2 (18 aa), loop 6 (99 aa), loop 8 (5 aa) and loop 12 (6 aa), along with an extended non-canonical neck-linker; there is also a ~ 40 aa insert between this neck linker region and MKLP2's coiled coil. However, the effects of these modifications on motor structure and function are unknown.

Visualisation of the kinesin-MT interaction and its sensitivity to nucleotide provides a structural framework for understanding motor functional output. Cryo-EM is an essential structural method for characterising kinesin-MT complexes and has, to date, revealed a mechanistic consensus for plus-end directed kinesins (*Atherton et al., 2014*; *Shang et al., 2014*; *Goulet et al., 2014*). To address the deficit in molecular understanding of the MKLP2 mechanism, we have determined cryo-EM reconstructions and calculated atomic models of MT-bound MKLP2 in different nucleotide states. These structures provide a visualisation of the conformations that functional MKLP2 undergoes, and which are blocked by small molecule inhibitors (*Tcherniuk et al., 2010*; *Labrière et al., 2016*). These structures, supported by biochemical and biophysical characterisations, reveal many atypical properties and behaviours of the MKLP2 motor domain. Strikingly, these noncanonical properties are superimposed on more generally conserved ATP-dependent subdomain movements within the

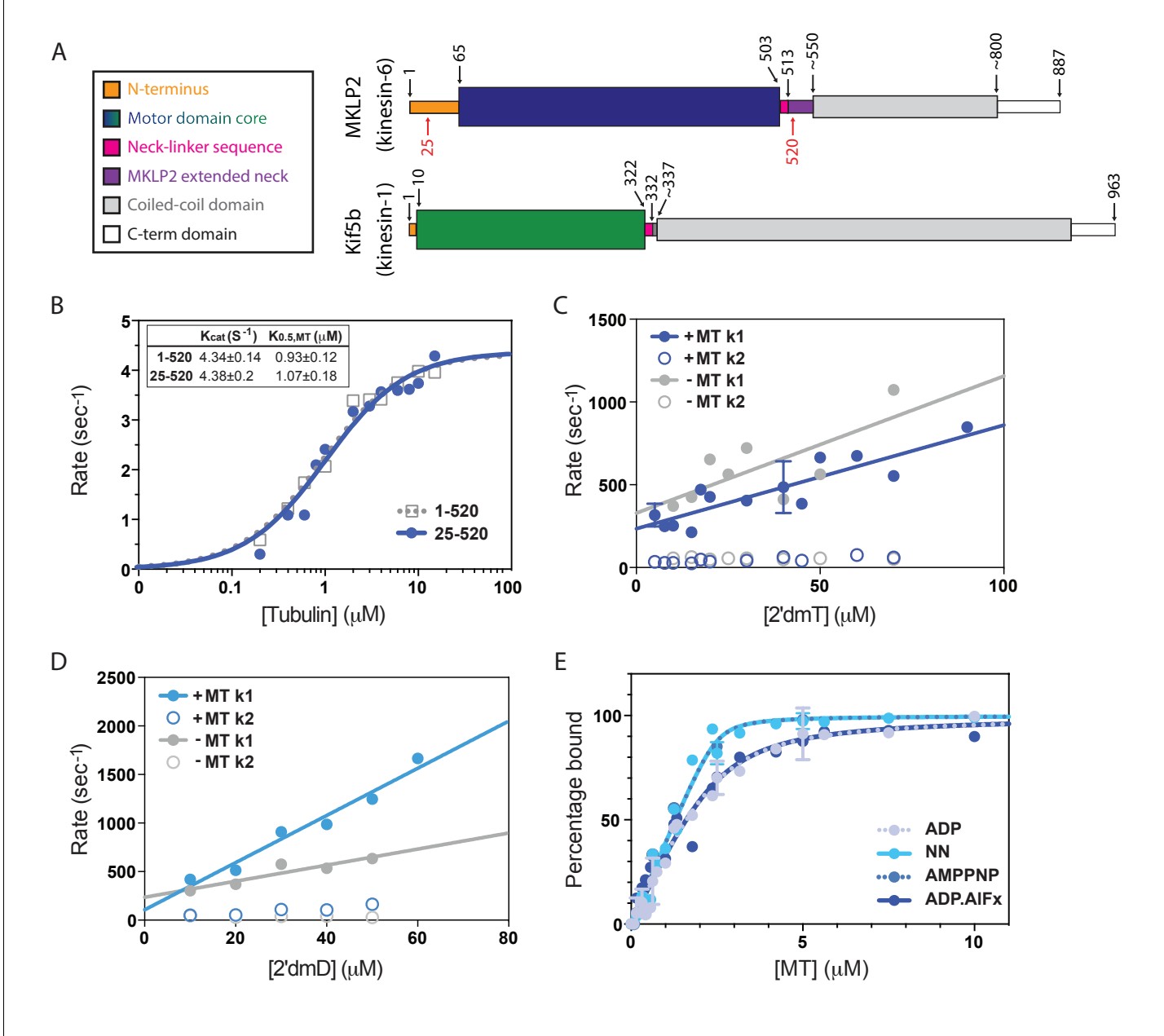

**Figure 1.** Atypical biochemistry of MKLP2 motor domain. (A). Domain organisation of MKLP2 and Kif5b (Kin1). Residues at domain boundaries are shown; the construct characterised in this study is indicated by red numbers. The motor domains are coloured in dark blue and green for MKLP2 and Kin1, respectively, and this colour scheme is used throughout. (B) MT-activated steady state ATPase velocity plotted as a function of [tubulin] for MKLP2 1–520 (grey) or 25–520 (blue). Data were fit to a Michaelis Menten kinetic (dashed grey, solid blue curves) yielding values for $k_{cat}$ and $K_{0.5,MT}$ shown (inset). (C) Kinetics of binding of 2' deoxy 3' mant ATP (2'dmT) to MKLP2-MD (25-520). The fluorescence increase consists of a fast (solid circles) and slow (open circles) phase. The rate constants for the two phases for a complex of MKLP2 with MTs (blue) showed a dependence on [2'dmT] that is very similar to those in the absence of MTs (grey). In both the presence/absence of MTs, the rate constant for the faster phase varied linearly with [2'dmT], defining apparent second order rate constants and y intercepts that are very similar to each other. (D) Kinetics of binding of 2' deoxy mant ADP (2'dmD) to MKLP2-MD (25-520). As in panel C, nucleotide free MKLP2 (grey) and nucleotide free MKLP2:MT (blue) were mixed with 2'dmD. (E) MT binding affinities of MKLP2-MD with different nucleotides. $K_d$s were determined by fitting to a quadratic equation. Two technical replicates for 7 data points of the NN- and ADP- states were performed. Where technical replicates were performed, the average and SEM are plotted, otherwise individual data points are shown.

The online version of this article includes the following figure supplement(s) for figure 1:

**Figure supplement 1.** The rate of ADP release from MKLP2-MD in the absence of MTs is unusually high.

motor domain. MKLP2 motor domain binding on the MT surface is shifted compared with other kinesins, and additional contacts between the MT and MKLP2 insertions in loops 2, 8 and 12 are formed. Additionally, while the MKLP2 neck-linker is directed towards the MT plus end in an ATP-like state enabling formation of the cover-neck bundle, the neck-linker does not fully dock against the motor core and exhibits more flexibility than is seen in transport motors such as Kin1 and kinesin-3 (Kin3). Moreover, the characteristic kinesin-6 large insertion in loop6 forms a discrete additional subdomain protruding from the MKLP2 motor core and away from the MT. Our characterisation provides a context in which the functional contribution of this motor in mitosis can be evaluated and offers a mechanistic framework in which other atypical members of the kinesin superfamily can be considered.

## Results

### The biochemical properties of the MKLP2 motor domain differ from those of other kinesins

We expressed monomeric mammalian MKLP2 motor domain constructs 1–520 and 25–520 and measured their steady-state MT-stimulated ATPase activity (*Figure 1B*). Fits of the data to a Michaelis Menten function reveal values of $k_{cat}$ = 4.34 ± 0.14 s$^{-1}$ and $K_{0.5,MT}$ = 0.93 ± 0.12µM for 1–520 and $k_{cat}$ = 4.38 ± 0.20 s$^{-1}$ and $K_{0.5,MT}$ = 1.07 ± 0.18µM for 25–520, showing that the presence or absence of the extended MKLP2 N-terminus does not affect steady state ATPase parameters. MT binding stimulates $k_{cat}$ >1000 fold (*Woehlke et al., 1997*; *Hackney, 1988*; *Gilbert et al., 1995*) to levels similar to the mitotic kinesin-5 (Kin5) and about 10-fold slower than the transport kinesins Kin1 or Kin3 (*Cochran et al., 2004*) (*Table 1*). Since the residues 1–25 are intrinsically disordered we used the 25–520 construct - which has a better yield during recombinant protein expression - for all subsequent assays. We refer to this 25–520 construct as MKLP2-MD.

To further dissect the MKLP2-MD ATPase cycle and its dependence on MTs, the dynamic interaction between MKLP2-MD and either 2'deoxy 3'mant ATP (2'dmT) or 2'deoxy 3'mant ADP (2'dmD) in the presence and absence of MT was measured (*Figure 1C,D*). In each case, the nucleotide binding transients consisted of fast and slow phases. The faster phase varies linearly with [nucleotide], implying that this is a readout of the initial binding of nucleotide to the active site; the slower phase shows little dependence on [nucleotide], implying that it measures a subsequent isomerization step. For

**Table 1.** Kinetic and affinity data of monomeric Kinesin-1,–3, and −5 compared to MKLP2-MD.

A summary of steady state ATPase activities and MT affinities of monomeric Kin1 (*Atherton et al., 2014*; *Woehlke et al., 1997*; *Hackney, 1988*; *Gilbert et al., 1995*; *Ma and Taylor, 1995*; *Rosenfeld et al., 1996*; *Nitta et al., 2004*; *Gigant et al., 2013*), whose structures were used to compared with MKLP2, as well as Kin3 (*Nitta et al., 2004*). Kin5's MT-stimulated ATPase activity has similar values as MKLP2 (*Cochran et al., 2004*; *Cochran et al., 2006*).

| Monomeric motor | MT-stimulated ATPase | | MT affinity; $K_d$ (µM) | | | |
|---|---|---|---|---|---|---|
| | $K_{0.5, MT}$ (µM) | $k_{cat}$ (S$^{-1}$) | *ADP* | *NN* | *ADP.AlFx* | *AMPPNP* |
| MKLP2-MD | 1.07 ± 0.18 | 4.38 ± 0.20 | 0.363 ± 0.057 | 0.043 ± 0.002 | 0.355 ± 0.073 | 0.043 ± 0.035 |
| Kin1 (Kif5a/b) | 12.7 ± 4.0 (*Atherton et al., 2014*) 26.0 ± 5.8 (*Rosenfeld et al., 1996*) | 34.2 ± 5.7 (*Atherton et al., 2014*) 43.6 ± 7.8 (*Rosenfeld et al., 1996*) 50.3 ± 1.6 (26) | 20.8 ± 2.4 (*Rosenfeld et al., 1996*) 2.0 ± 0.3 (*Gigant et al., 2013*) | n.a. | 1.4 ± 0.2 (*Rosenfeld et al., 1996*) 0.3 ± 0.006 (*Gigant et al., 2013*) | 1.1 ± 0.1 (*Rosenfeld et al., 1996*) 0.045 ± 0.012 (*Gigant et al., 2013*) |
| Kin3 (Kif1a) | 0.0537 ± 0.0057 (*Atherton et al., 2014*) | 43.4 ± 1.0 (*Atherton et al., 2014*) | 0.0068 ± 0.0025 (*Nitta et al., 2004*) | n.a. | 0.0059 ± 0.0015 (*Nitta et al., 2004*) | 0.0042 ± 0.0013 (*Nitta et al., 2004*) |
| Kin5 (Eg5) | 4.5 (*Cochran et al., 2004*) 0.29 ± 0.02 (*Cochran et al., 2006*) | 2.9 (*Cochran et al., 2004*) 5.5 ± 0.1 (*Cochran et al., 2006*) | n.a. | n.a. | n.a. | n.a. |

n.a.: not available.

2'dmT (*Figure 1C*), rates in the presence and absence of MT are nearly identical (including Y intercept), implying that – unlike e.g. Kin-1 or Kin-5 – the kinetics of binding 2'dmT to MKLP2-MD are essentially independent of MTs. Binding of 2'dmD produces similar behaviour (*Figure 1D*), with an approximately two-fold increase in the apparent second order rate constant for 2'dmD binding. Extrapolation of the linear fit to the origin defines an apparent 2'dmD dissociation rate constant, and in other kinesins, such plots typically go through the origin in the absence of MTs, implying tight ADP binding within the error of the measurement (*Hackney, 1988*; *Gilbert et al., 1995*; *Cochran et al., 2004*; *Ma and Taylor, 1995*). However, as is evident from *Figure 1D*, the corresponding plot for MKLP2-MD intercepts the y axis well above the origin. Although this is an indirect way of calculating the ADP dissociation rate constant, it implies that the interaction of ADP with MKLP2 is unlike that of other kinesins. In order to directly examine the kinetics of ADP release from MKLP2-MD in the absence of MTs, we formed a complex of MKLP2-MD:2'dmD and mixed it in the stopped flow with a large excess of unlabelled ADP. *Figure 1—figure supplement1* shows that the resulting fluorescence decrease, which reflects 2'dmD dissociation, fits a single exponential process with rate constant of $51.9 \pm 0.1$ s$^{-1}$, i.e. the rate of ADP release from MKLP2-MD in the absence of MTs is unusually high. Taken together, these data suggest that there is a relative uncoupling of allosteric communication between the MT binding and catalytic sites in MKLP2-MD compared to other kinesins that have been described (*Cochran et al., 2004*; *Ma and Taylor, 1995*).

The binding affinity of MKLP2-MD to MTs was measured as a function of nucleotide state by a co-sedimentation assay, using a fixed [MKLP2-MD] and a range of [MT]. Plots of fraction of bound MKLP2-MD versus [MT] are depicted in *Figure 1E*. These were fit to a quadratic function assuming a 1:1 motor:tubulin dimer stoichiometry to yield apparent dissociation constants (*Figure 1E*; *Table 1*). Like other kinesin family members (*Table 1*) (*Atherton et al., 2014*; *Cochran et al., 2004*; *Rosenfeld et al., 1996*; *Nitta et al., 2004*; *Gigant et al., 2013*; *Cochran et al., 2006*), MT affinity of MKLP2-MD is modulated by the state of the catalytic site, with binding in the absence of nucleotide (NN) or presence of AMPPNP demonstrating a higher affinity than that with ADP or ADP.AlFx (*Table 1*). In marked contrast to other kinesins, however, the ADP state of MKLP2-MD - which usually represents the kinesin weak binding state (*Rosenfeld et al., 1996*; *Gigant et al., 2013*; *Crevel et al., 1996*) – demonstrates a relatively high affinity. Thus, the difference in affinity between nucleotide states that in other kinesins are 'strong' binding (NN, AMPPNP) and 'weak' binding (ADP) is only about 8.5-fold. Overall, MKLP2-MD associates with MTs relatively strongly throughout its ATPase cycle. Together, these results provide additional evidence of the non-canonical properties of the MKLP2-MD mechanism.

## Conserved nucleotide-dependent subdomain rearrangements provides a structural framework for MKLP2 divergence

To visualise the MKLP2-MD (*Figure 2A*) and provide insight into these unusual biochemical properties, we calculated MT-bound MKLP2-MD reconstructions with different nucleotides bound: (1) ADP; (2) no nucleotide (NN) using apyrase treatment; (3, 4) ATP-like analogues (AMPPNP and ADP.AlFx) (*Figure 2—figure supplement 1*). Our ability to determine an ADP-bound structure highlights the tighter association of this motor with MTs throughout its ATPase cycle and is consistent with the biochemical data (*Figure 1E*). The asymmetric unit of all these reconstructions is the MKLP2 motor domain bound to an αβ-tubulin dimer within the MT (*Figure 2B*), which shows a resolution gradient between the MT and bound MKLP2 (*Figure 2—figure supplement 1*; *Figure 2—figure supplement 2*); for interpretation of MKLP2-MD mechanochemistry, reconstructions were filtered according to the local resolution of the motor domain. To facilitate interpretation of the cryo-EM density, we calculated comparative models for each nucleotide state and docked them into the density using flexible fitting (*Table 2*, *Figure 2—figure supplement 3*, *Figure 2—figure supplement 4*). In addition, multiple loop conformations were generated and clustered to represent loops with ambiguous fit. This yielded a set of models (*Table 2*; *Figure 2—figure supplement 4*) that show nucleotide-dependent conformational changes of MT-bound MKLP2 including the conformation and variation of the MKLP2-specific inserts (*Figure 2A*).

These structures show that the overall organisation of MKLP2-MD is similar compared to other kinesins, with the major contact site between motor and MT being centred over the tubulin intradimer interface (*Figure 2B*). The coordinated conformational changes that occur in Kin1 in response to MT and nucleotide binding have been described in terms of three independently articulated

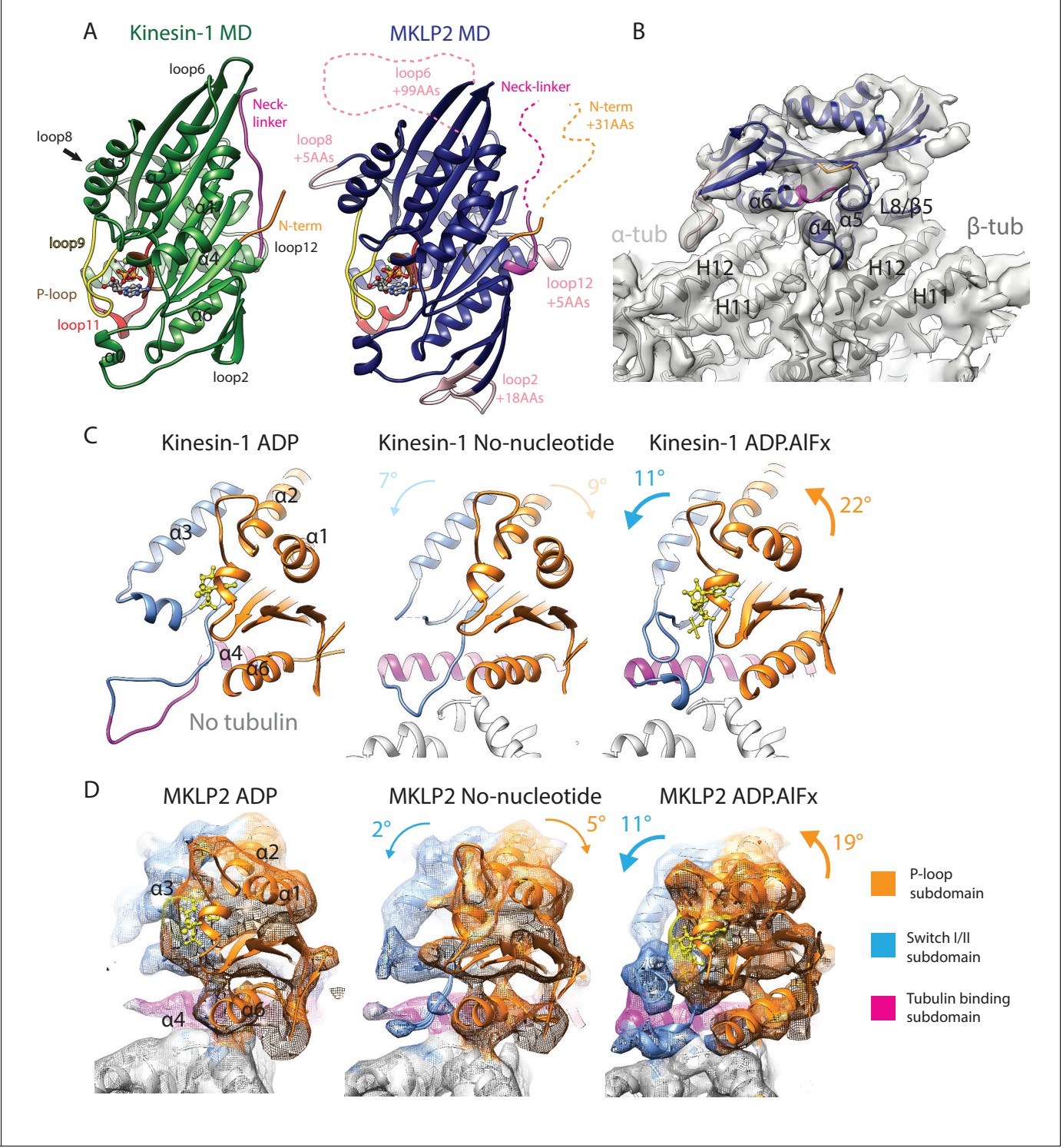

**Figure 2.** MKLP2 motor domain, cryo-EM reconstructions and nucleotide-dependent subdomain movements. (**A**). The MKLP2-MD has multiple loop inserts compared to Kin1. The Kin1 MD-ADP.AlFx structure (PDB: 4HNA) is shown on the left. Our MKLP2-MD-ADP.AlFx model is shown on the right, with its specific loop inserts coloured and labelled in pink. The 25–520 construct contains 31 extra residues at its N-terminus compared to Kin1, while its C-terminus ends at an equivalent length. (**B**) The asymmetric unit of the MKLP2-MD-ADP.AlFx MT-bound cryo-EM reconstruction (grey transparent density) with docked MKLP2-MD model in blue, α-tubulin model in light grey and β-tubulin model in dark grey. MKLP2-MD and tubulin secondary structure elements at the interface are labelled. (**C and D**) Subdomain rearrangements in Kin1 crystal structures (**C**) and MKLP2-MD (**D**) viewed towards the MT plus end. Subdomains for kinesin-1 and MKLP2-MD are assigned and coloured based on Cao et al (*Cao et al., 2014*). (**C**) Kinesin-1: left, ADP

*Figure 2 continued on next page*

*Figure 2 continued*

(PDB: 1BG2); middle, NN + tubulin (PDB: 4LNU); right ADP.AlFx +tubulin (PDB: 4HNA). (**D**) The MKLP2-MD models are shown within cryo-EM density also coloured by subdomain. For each nucleotide-dependent transition, subdomain rotations for the P-loop and SwitchI/II subdomains are shown in degrees relative to the tubulin binding subdomain, which remains static in all states. Subdomain movements between Kin1 ADP and NN states (indicated in light text) are not directly comparable to their MKLP2-MD counterparts, because the Kin1-ADP structure is not tubulin-bound and has an atypical tubulin-binding subdomain conformation.

The online version of this article includes the following figure supplement(s) for figure 2:

**Figure supplement 1.** Evaluation of the resolutions and nucleotide occupancy of the MKLP2-MD-MT reconstructions.
**Figure supplement 2.** High-resolution tubulin density features.
**Figure supplement 3.** A protocol describing the different stages for creating atomic models of MKLP2-MD.
**Figure supplement 4.** Local Fitting Improves at Each Model Refinement Stage.

subdomains that move with respect to each other during the MT-bound ATPase cycle (*Shang et al., 2014*; *Cao et al., 2014*): the tubulin-binding subdomain, the Switch I/II subdomain and the P-loop subdomain (*Figure 2C*). This structural simplification is useful in describing global rearrangements in the motor domain. Our reconstructions demonstrate that the overall features of nucleotide-dependent subdomain movement described for Kin1 also apply to MKLP2 (*Figure 2D*; *Video 1*), albeit the comparison is incomplete given the lack of a tubulin/MT-bound Kin1 ADP structure. In the transition from ADP-bound to NN structures (*Figure 2D*, left to middle), the MKLP2 Switch I/II and P-loop subdomains both rotate away from each other slightly and relative to the static tubulin-binding domain. In the transition from the NN to ADP.AlFx conformations (*Figure 2D*, middle to right), both domains rotate towards the catalytic site, with the rotation of the P-loop subdomain being greater than the Switch I/II subdomain. This supports the idea that these subdomain movements are a conserved facet of kinesin mechanochemistry even amongst motors with highly distinct functions. To determine how MKLP2's divergent mechanochemistry functionally harnesses these subdomain movements, and

**Table 2.** MKLP2 Homology modelling and validation.

Columns 2–3 show the list of PDB IDs that were used as templates for specific structural regions of each MKLP2 nucleotide state (4LNU (*Cao et al., 2014*), 4HNA (*Gigant et al., 2013*), 4OZQ (*Arora et al., 2014*), 1VFV (*Nitta et al., 2004*), 3GBJ and 4Y05 are structures from the Structural Genomics Consortium). Columns 4–5 show the overall QMEAN scores calculated before, and after refinement and show that model quality is maintained following refinement. Columns 6–7 show the global cross-correlation scores before and after refinement.

| | Templates used for homology modelling | | QMEAN scores | | Global Cross-correlation scores | |
|---|---|---|---|---|---|---|
| **Nucleotide state** | *MKLP2 Structure* | *Template* | *Homology Model* | *Refined Model* | *Homology Model* | *Refined Model* |
| *ADP* | Main template | 3GBJ | 0.636 | 0.623 | 0.908 | 0.931 |
| | Helix α4, helix α5, loop 8 | 4LNU | | | | |
| | Helix α4/loop 12 | 4OZQ | | | | |
| | Loop 2 | 4Y05 | | | | |
| *NN* | Main template | 4LNU | 0.669 | 0.663 | 0.913 | 0.930 |
| | Helix α4/loop 12 | 4OZQ | | | | |
| | Loop 2 | 4Y05 | | | | |
| | Loop 5 | 3GBJ | | | | |
| *ADP.AlFx* | Main template | 4HNA | 0.671 | 0.689 | 0.860 | 0.890 |
| | Helix α4/loop 12 | 4OZQ | | | | |
| | Loop 2 | 4Y05 | | | | |
| | Loop 5 | 3GBJ | | | | |
| *AMPPNP* | Main template | 1VFV | 0.658 | 0.679 | 0.925 | 0.943 |
| | Helix α4, helix α5, loop 8 | 4LNU | | | | |
| | Helix α4/loop 12 | 4OZQ | | | | |
| | Loop 2 | 4Y05 | | | | |

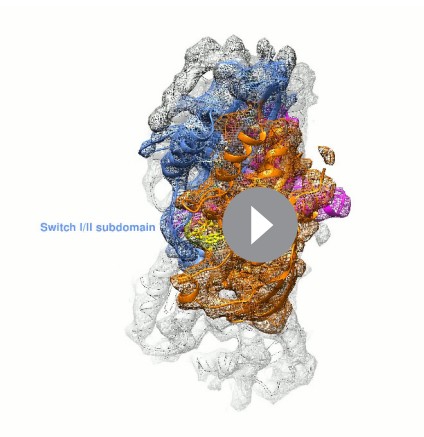

**Video 1.** Subdomains within MKLP2-MD-ADP.AlFx cryo-EM reconstruction. The organisation of kinesin subdomains is illustrated within the MKLP2-MD-ADP.AlFx model, with cryo-EM density shown as colored mesh and the model as colored ribbon (Switch I/II subdomain, blue, P-loop subdomain, orange, Tubulin-binding subdomain magenta, α-tubulin, light grey and β-tubulin, dark grey). ADP.AlFx nucleotide is shown as yellow ball-and-stick representation; nucleotides and paclitaxel have been removed from the tubulin model for clarity. Also for clarity, density mesh for loop6 is not colored. A clipping plane is moved though the MD to

to further characterise MKLP2's differences compared to other kinesins, we examined the response of different regions of the MT-bound MKLP2-MD in different nucleotide states.

## Atypical response of MKLP2 catalytic site to nucleotide

At the MKLP2-MD catalytic site, the nucleotide-dependent movement of three conserved, mobile elements that lie at the junction of the three subdomains can be tracked: (1) the P-loop (brown); (2) loop9 (yellow, contains Switch I); (3) loop 11 (red, contains Switch II) (*Figure 3*, *Video 2*). The N-terminal half of helix-α4 lies at the back of the nucleotide-binding site and provides a structural link to the MT binding interface. These elements are sensors of the nucleotide present in the active site and also participate in the conformational response of the motor domain to nucleotide.

The conformation(s) of MT-bound kinesins in the presence of ADP are still relatively poorly described (*Atherton et al., 2014*) especially at high resolution, and depend on the MT affinity and kinetic behaviour of individual proteins. The properties of MKLP2-MD are such that an MT-

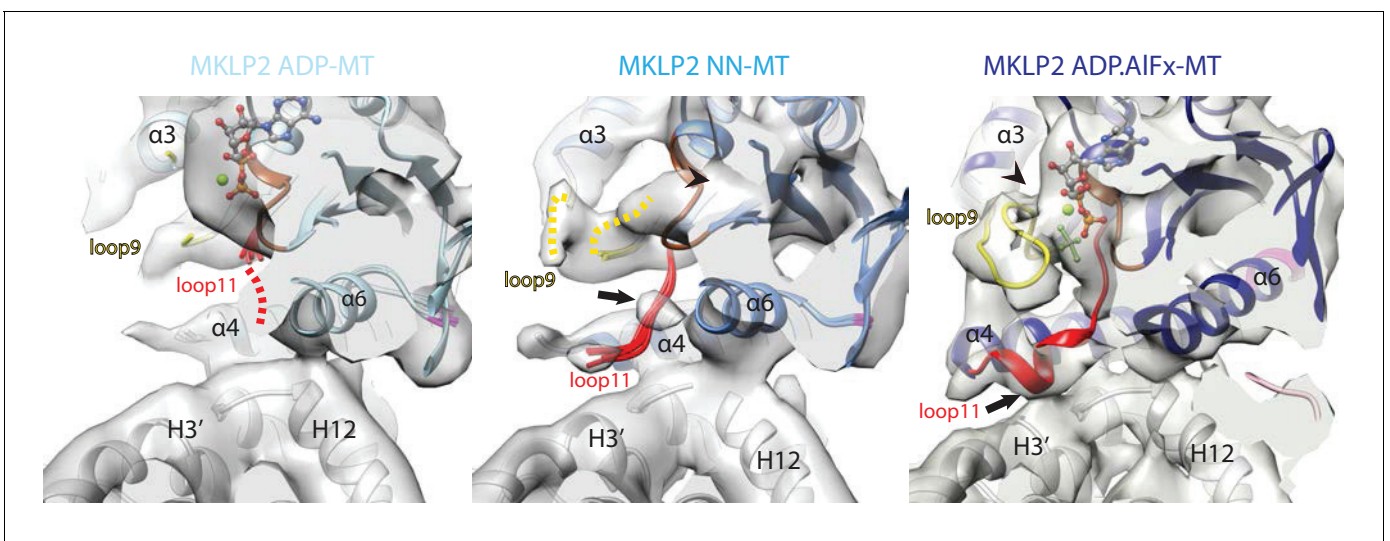

**Figure 3.** Conservation and divergence at the MKLP2-MD nucleotide binding pocket. View of the MKLP2-MD nucleotide binding pocket showing nucleotide-dependent transitions in helix-α4, P-loop (brown), loop9 (yellow) and loop11 (red). In the ADP state (left), density corresponding to ADP is visible and connected to the P-loop, while in the NN state (middle), density for nucleotide is no longer present (arrowhead). In the ADP.AlFx state, a hydrolysis-competent 'closed' nucleotide pocket conformation is observed, loop11 becomes fully ordered with a single helical turn contacting α–tubulin H3' (arrow), and loop9 forms an ordered β-hairpin which contacts the nucleotide (arrowhead).

The online version of this article includes the following figure supplement(s) for figure 3:

**Figure supplement 1.** Comparison of MKLP2 nucleotide binding site with other N-kinesins.

**Figure supplement 2.** Conservation of nucleotide-dependent subdomain movements and location and configuration of the clefts that separate them in Kin1 and MKLP2.

**Figure supplement 3.** MT-bound MKLP2-MD-AMPPNP does not adopt an 'ATP-like state'.

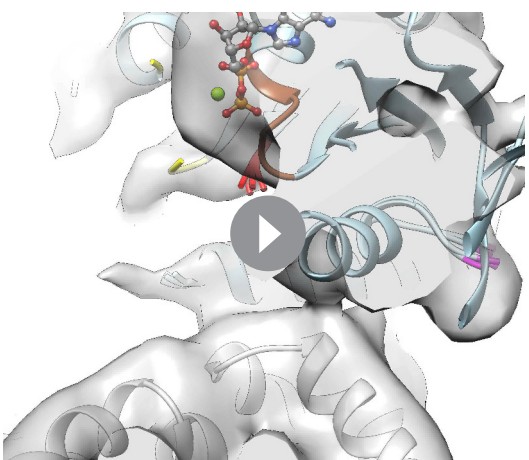

**Video 2.** Changes at the nucleotide-binding site during the MKLP2-MD ATPase cycle. Nucleotide-state transitions are represented by morphing cryo-EM density between MKLP2-MD-ADP and MKLP2-MD-NN reconstructions then MKLP2-MD-NN and MKLP2-MD-ADP.AlFx reconstructions using Chimera's (*Goddard et al., 2007*) 'Morph Map' tool (with the option of adjusting threshold for a constant volume). The MKLP2-MD-ADP fitted model is shown first and substituted for the MKLP2-MD-NN fitted model halfway through the ADP to NN density morph. The MKLP2-MD-NN fitted model is then substituted for the MKLP2-MD-ADP.AlFx fitted model halfway through the ADP to NN density morph.

https://elifesciences.org/articles/27793#video2

bound MKLP2-MD-ADP structure could be captured, in which density corresponding to ADP is coordinated primarily via the P-loop with some connectivity to Switch II/loop11 (*Figure 3*, left; *Figure 2—figure supplement 1*). A portion of loop11 is visible near the P-loop and the N-terminus of helix-α6, while most is not visible. This flexibility in loop11 is also seen in Kin1-ADP (*Kull et al., 1996*) and MT-bound Kin3-ADP (*Atherton et al., 2014*). However, in MT-bound MKLP2-MD-ADP, loop9 as well as the N-terminus of helix-α4 are more flexible compared to these other kinesin-ADP conformations, reflecting distinct properties of MKLP2-MD.

Highly unusually, in the MKLP2-MD-NN structure, no major conformational differences are seen compared to MKLP2-MD-ADP (*Figure 3*, middle), although nucleotide density is clearly lacking from the active site (*Figure 2—figure supplement 1*). Local differences due to subtle rotations in both the Switch I/II and the P-loop subdomains are seen (*Figure 2D*, middle), and helix-α4 is longer by ~1 turn compared to MKLP2-MD-ADP, but its N-terminal end along with most of loop11 are flexible. However, unlike in the ADP-bound structure, a region of Switch II density in loop11 now appears connected to helix-α4. The connection to helix-α4 is close to the highly conserved 'linchpin' Asn (*Shang et al., 2014*) (N430 in mouse MKLP2, N255 in Kin1), is also seen in Kin1/3 NN states (*Atherton et al., 2014*; *Shang et al., 2014*; *Cao et al., 2014*). In addition, density corresponding to the P-loop has shifted away from the MT surface, and while density corresponding to loop9 is visible, with part of it contacting the P-loop (*Figure 3*, middle), loop9 does not adopt a defined conformation, due to flexibility. The overall disordered nature of the N-terminus of helix-α4, loop 11 and loop 9 is very different from the more ordered conformation of these elements consistently seen in the MT-bound NN state of other plus-end kinesins (*Figure 3—figure supplement 1*)(*Atherton et al., 2014*; *Shang et al., 2014*; *Cao et al., 2014*; *Goulet et al., 2012*). Moreover, the well-defined connectivity of loop11 and the MT surface seen in Kin1/3/5 NN states is not well ordered in MKLP2-MD. Thus, the MKLP2-MD-ADP and –NN structures are more similar to each other with respect to flexibility of MT binding elements at the helix-α4 N-terminus and loop11 than is typical for other kinesins.

Transition to the MKLP2-MD-ADP.AlFx structure causes substantial local rearrangements in the vicinity of the active site (*Figure 3*, right), which are in turn linked to large subdomain rearrangements compared to the ADP-to-NN transition (*Figure 2D*). These rearrangements can also be described by the configuration of clefts between the subdomains (*Shang et al., 2014*); *Figure 3—figure supplement 2*, see also Discussion). First, density connects the P-loop and loop11 in the presence of bound nucleotide (*Figure 3*, right). However, in the presence of the ADP.Pi-like analogue, stable conformations for all components around the active site are also observed: density corresponding to the N-terminal end of helix-α4 and loop11 are seen and exhibit connectivity to adjacent regions (*Figure 3*, right), while the N-terminus of loop11 runs past the P-loop, the γ-phosphate mimic, and is connected to helix-α4, (*Figure 3*, right). The C-terminal end of loop11 forms a helical turn and density in this region connects to the N-terminus of helix-α6. There is also connectivity between the N-terminal end of helix-α4 and loop9, while the proximity of loop9 and loop11 is consistent with a 'phosphate tube' structure as seen in other ATP/ADP.Pi-like kinesin structures (*Gigant et al., 2013*; *Sindelar and Downing, 2010*; *Chang et al., 2013*; *Parke et al., 2010*). Overall,

the compactness of the ATPase site is consistent with adoption of a catalytically competent conformation (*Parke et al., 2010*), similar to that seen in other kinesins, highlighting mechanochemical conservation within the superfamily with respect to closure of the active site to enable ATP hydrolysis.

However, the MKLP2-MD-AMPPNP reconstruction illustrates another interesting divergence of MKLP2-MD mechanochemistry compared to Kin1/3 (*Figure 3—figure supplement 3*). In Kin1 and Kin3, the motor conformation in the presence of AMPPNP and ADP.AlFx are essentially indistinguishable (*Atherton et al., 2014*; *Gigant et al., 2013*). However, in MKLP2-MD, while density corresponding to bound AMPPNP is observed (*Figure 2—figure supplement 1*), none of the characteristic features in the motor observed for the catalytically competent ADP.AlFx conformation are seen upon AMPPNP binding. Instead, this reconstruction exhibits a similar conformation to the MKLP2-ADP and –NN reconstructions, such that at the nucleotide binding site, helix-α4 shows a partially extended conformation and only some density corresponding to loop9 and loop11 is visible (*Figure 3—figure supplement 3B*). This reconstruction thereby illustrates the large difference in structural response of MKLP2-MD to different ATP-like analogues - AMPPNP compared to ADP.AlFx – and is a further readout of the divergent properties of MKLP2.

## MKLP2 has a non-canonical MT binding interface and footprint on the MT surface

The divergent properties of MKLP2-MD are further emphasised in its interaction with the MT surface. The MKLP2-MD interface with the MT is composed of nucleotide-invariant and nucleotide-sensitive elements. The nucleotide-invariant elements are formed by the tubulin-binding subdomain (*Figure 2C,D*). In this subdomain of MKLP2-MD, a 5 aa insertion (*Figure 2A*) both contributes an additional helical turn to the C-terminus of helix-α4 and adds length to loop12 compared to Kin1 (*Figure 4A*); however this insert does not contact the MT. This C-terminus extension of helix-α4 is nucleotide-insensitive (*Figure 4—figure supplement 1A,D*) and thus is structurally similar to the helix-α4 extension formed by part of Kin3's loop12 insert (*Atherton et al., 2014*), although these insertions have no sequence homology.

MKLP2-MD subdomain rearrangement in response to nucleotide (*Figure 2D*) causes nucleotide-dependent conformational changes in other elements at the MT binding interface. This includes movement of loop7 in the Switch I/II subdomain closer to the MT in the ATP-like state (*Figure 4B*, *Figure 4—figure supplement 1B,E*), and movement of helix-α6 in the P-loop subdomain relative to the MT surface (*Figure 2D*). As described above, however, in the vicinity of the nucleotide binding site, MKLP2-MD loop11 and helix-α4's N-terminus are less ordered than in Kin1 in the absence of nucleotide (*Figure 3*, *Figure 3—figure supplement 1C,D,E*).

MKLP2's Switch I/II subdomain also contains a 5 aa insertion in loop8 (*Figure 2A*), which is only well ordered in the presence of ADP.AlFx, and reaches across to contact H10-β9 of the neighbouring protofilament within the MT (*Figure 4B*, *Figure 4—figure supplement 1B,E*). To our knowledge MKLP2-MD is the first kinesin described where a monomeric motor domain can bridge across two protofilaments. MKLP2-MD also has a larger 18 aa insert (*Figure 2A*) in both the β-sheet1 and loop2 of the P-loop subdomain. In the NN and ADP states, density for this insert is not visible unless much less conservative thresholds are used, indicating greater flexibility of loop2 in these two states (*Figure 4—figure supplement 1C,F*). In the presence of ADP.AlFx, density for this element extends from the minus end of the MKLP2-MD and connects to α-tubulin (*Figure 4C*), consistent with the large rotation of the P-loop subdomain in this ADP.AlFx state (*Figure 2D*).

Compared to Kin1, not only does the MKLP2-MD contain extra sequences that contact the MT, but the precise position and orientation of MKLP2-MD as a whole is also noticeably shifted and rotated on the MT surface (*Figure 4D*). A similar shift and rotation is present in both our MT-bound NN and ADP.AlFx MKLP2-MD reconstructions (*Figure 4—figure supplement 2*), and is therefore independent of the nucleotide-dependent conformational changes. This shift and rotation particularly alters the MT interface of MKLP2-MD near the nucleotide binding pocket. For example, in MKLP2-MD, loop7 is closer to β-tubulin (*Figure 4E*) compared to Kin1, while the ordered single-turn helix in loop11 is closer to H3 than H12 in α-tubulin in the MKLP2-MD-ADP.AlFx state (*Figure 4F*).

Altogether, both kinesin-conserved and MKLP2-MD-specific elements contribute to the motor's interaction with the MT, several of which are nucleotide sensitive. The collective effect of MKLP2-specific modifications leads to the shift and rotation of the MKLP2-MD on the MT surface. The differences in MKLP-MD compared to Kin1 also modify its footprint on the MT surface, leading to MKLP2-

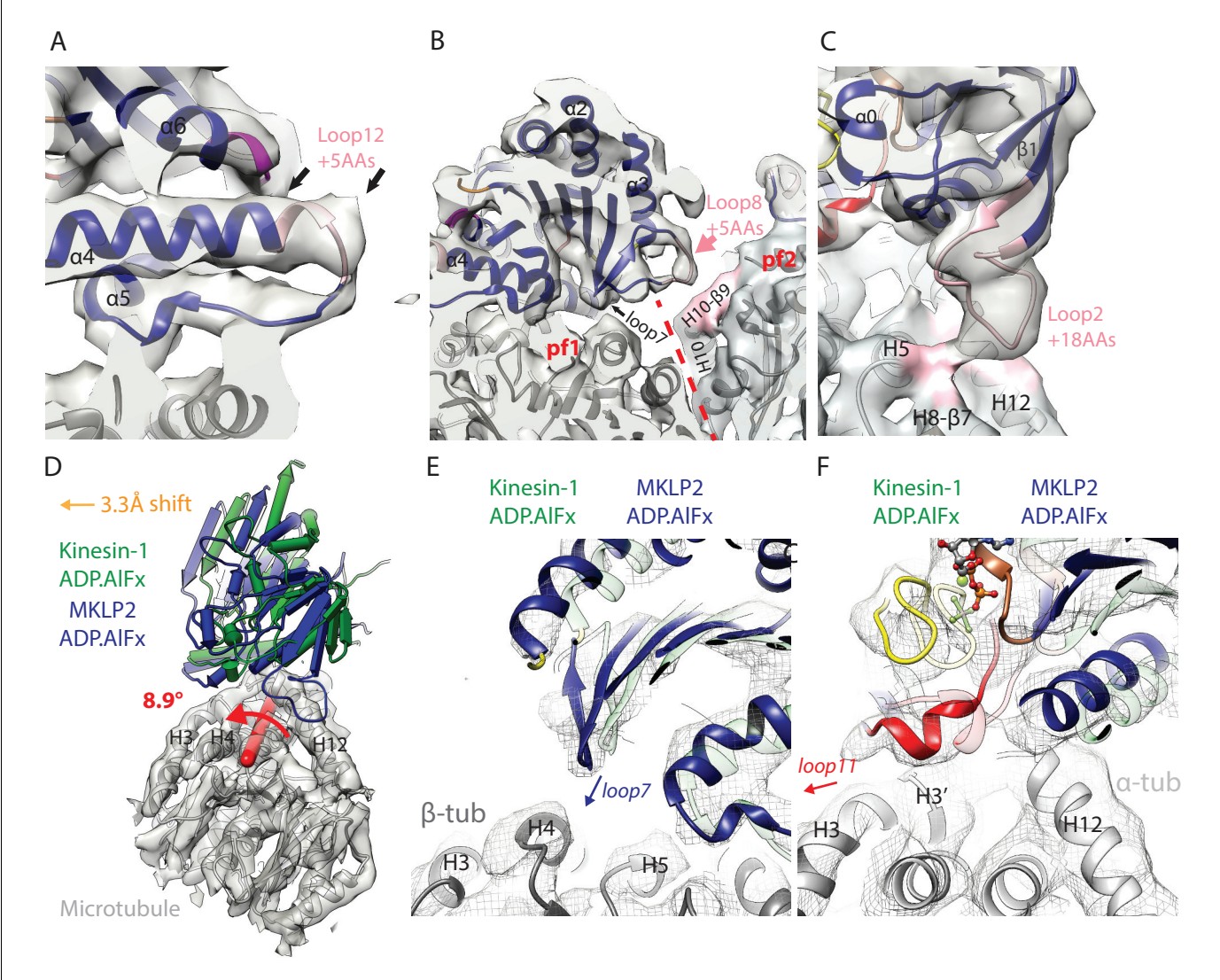

**Figure 4.** MKLP2-specific loop insertions at the MT interface and global tilt and shift of MKLP2-MD on the MT surface. (**A**). Loop12 insertion (pink) partially contributes to the C-terminal extension of helix-α4 (extra length indicated by arrows) but does not contact the MT. This extension is present in all nucleotide states; shown here for MKLP2-MD-ADP.AlFx. (**B**) In MKLP2-MD-ADP.AlFx the loop8 insertion (pink) reaches across to β-tubulin in the adjacent protofilament (pf, demarcated by a red dashed line). Tubulin density <7 Å away from the insertion is coloured in pink. (**C**) In MKLP2-MD-ADP. AlFx the loop2 insertion (pink) reaches towards the MT minus end contacting α-tubulin. Tubulin density <7 Å away from the insertion is coloured in pink. (**D**) Overlay of asymmetric unit MKLP2-MD-ADP.AlFx MT-bound model (blue pipes and planks) and Kin1 (green pipes and planks) MT-bound model (*Atherton et al., 2014*) when the tubulin dimers are superimposed. The density and model of tubulin in the MKLP2-MD-ADP.AlFx model is shown (transparent grey density). When their tubulins are aligned, a global shift (orange arrow indicates shift direction) and rotation (on the axis shown in red) of the MKLP2-MD on the MT surface is observed compared to Kin1. (**E**) This relative shift and rotation moves MKLP2-MD's (blue model) loop7 closer to β-tubulin. (**F**) MKLP2-MD Loop11 is further from α-tubulin H12 and closer to H3 in comparison to Kin1 (green transparent model). Density for MKLP2-MD-ADP.AlFx-MT (grey mesh).

The online version of this article includes the following figure supplement(s) for figure 4:

**Figure supplement 1.** MKLP2 specific inserts are nucleotide sensitive.

**Figure supplement 2.** MKLP2 has an altered binding orientation on the MT surface in all nucleotide states.

MD contacting a greater MT surface area in all states (*Figure 5*; *Figure 5—figure supplement 1*). The enhancing effect of the shift and rotation (*Figure 4D*) is most clearly seen in the increased footprint of loop7 and α4 (in both NN and ADP.AlFx states), and loop11 in the ADP.AlFx state (*Figure 5*, *Figure 5—figure supplement 1*). The transition from NN to ADP.AlFx states and further rotation of the Switch I/II subdomain brings loop7 and loop8 into even more extensive contact with β-tubulin's

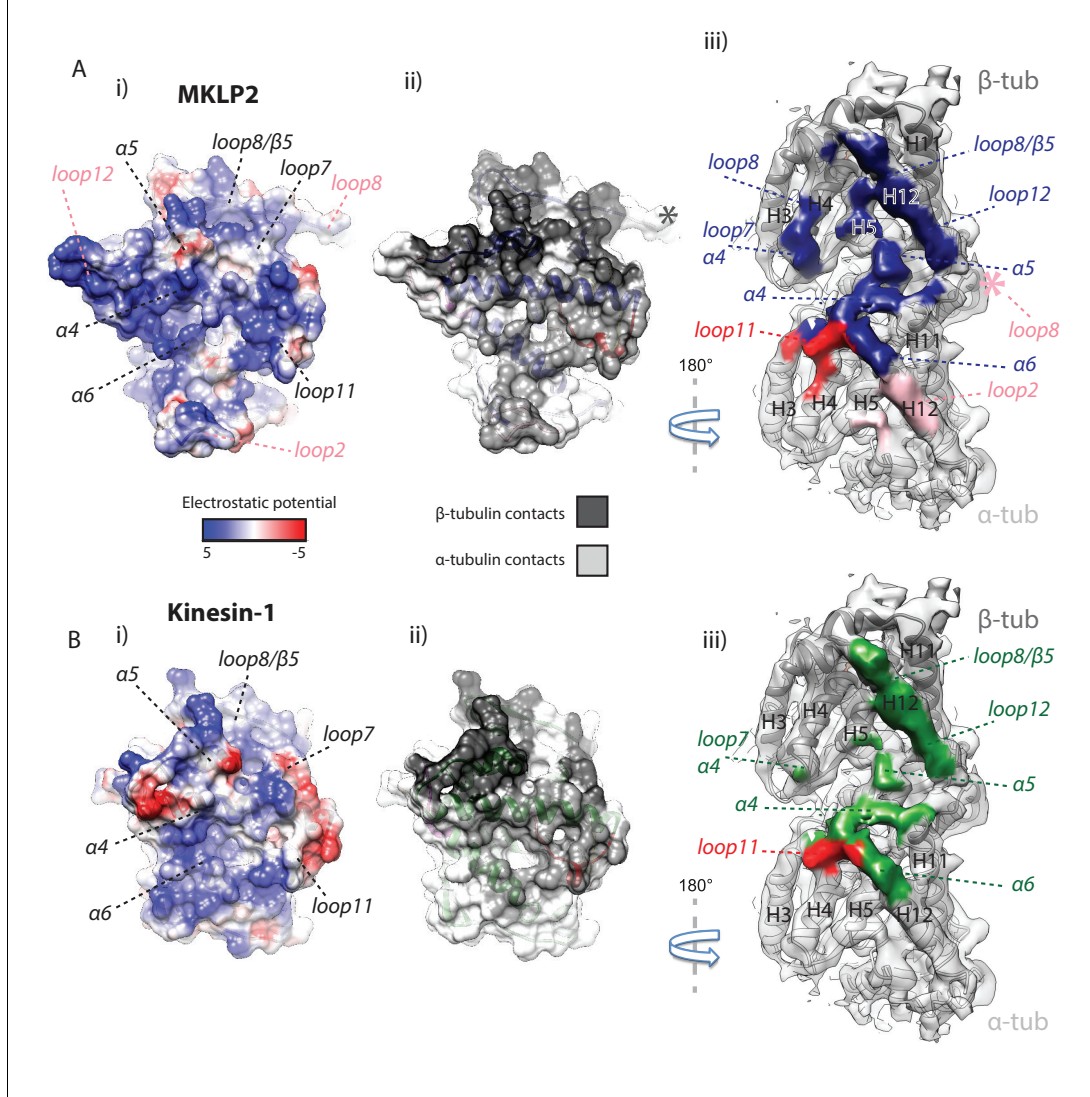

**Figure 5.** MKLP2 has an altered footprint on the MT surface. The binding interface of (**A**) MKLP2-MD-ADP.AlFx compared to (**B**) Kin1-MD-ADP.AlFx. (i) The binding surface of MKLP2-MD and Kin1 coloured according to the electrostatic potential. (ii) The binding surface of MKLP2-MD and Kin1 coloured according to α- or β-tubulin contacts. (iii) MT footprint of MKLP2-MD (top) compared to Kin1 (bottom). Coloured labels and dashed lines indicate contacting secondary structure elements in each motor. Tubulin density <7 Å distance from the bound motors is coloured. In MKLP2-MD-ADP.AlFx, loop2 is ordered and contacts α-tubulin (pink density) while the ordered loop8 contacts the neighbouring tubulin asymmetric unit (pink asterisk). The online version of this article includes the following figure supplement(s) for figure 5:

**Figure supplement 1.** MKLP2 has an altered footprint on the MT surface in all nucleotide states.

H4, while full ordering of helix-α4 and loop11 adjacent to the active site leads to additional contacts with α-tubulin. Moreover, ordering of MKLP2-specfic insertions that are present in the ADP.AlFx reconstruction produce a further enhancement of the MKLP2-MD footprint (*Figure 5*). In addition to the enlarged footprint of MKLP2-MD on the negatively charged MT surface (*Woehlke et al., 1997*), the electrostatic surface of MKLP2-MD shows a more pronounced positive interaction surface compared to kinesin-1, especially around the C-terminal end of helix-α4, a region that is nucleotide invariant. Thus, through its ATPase cycle, the footprint of MKLP2-MD on the MT is altered and is most different in its ATP-like state.

## Divergence of MKLP2 motor domain structural response to nucleotide: neck-linker and N-terminus

Nucleotide-dependent subdomain rearrangements in MKLP2-MD yield conformational changes at both termini of the motor. This includes opening of the cleft between the P-loop and tubulin-binding subdomains that can accommodate neck-linker reorientation and docking upon ATP binding (*Figure 6*). With ADP bound, as for other kinesins, the C-terminal helix-α6 lies close to the motor-MT interface and terminates adjacent to the C-terminus of helix-α4 (*Figure 6A*, arrow). The density for helix-α6 is less well-defined, with a length of three helical turns at most and is thus shorter than typically seen in other kinesins (*Figure 6A*, arrowhead; [*Atherton et al., 2014*; *Shang et al., 2014*]). In this configuration, extension of helix-α6, docking of the neck-linker and formation of the cover-neck bundle (CNB) are all prevented. Strong density corresponding to only a few residues of each of the N- and C-termini is observed, although additional density alongside β-sheet1 is visible suggesting partial occupancy of neck-linker conformers directed towards the MT minus end (*Figure 6A*, dotted line). In the NN reconstruction, the overall organisation of this region is very similar to that in the ADP state (*Figure 6B*) (and of the AMPPNP state, *Figure 6—figure supplement 1*), although density corresponding to helix-α6 is better defined in the NN compared to the ADP structure.

Following the subdomain rearrangement accompanying the NN/ADP.AlFx transition, the neck-linker cleft opens. Helix-α6 of MKLP2-MD extends beneath the core β-sheet, thereby directing the neck-linker towards the MT plus end (*Figure 6C*). This is consistent with the presence of a conserved cluster of hydrophobic residues in helix-α4, helix-α6 and the N-terminus that form a structural pocket that accommodates the helix-α6 extension and neck-linker reorientation (*Figure 6D*). This is also consistent with orientation of the N-terminus towards the MT plus end and formation of the CNB. The density of these N-terminal residues corresponds to S61-V64 and for the neck-linker corresponds to Q505-H508. In support of partial docking and reorientation of the neck-linker, anisotropy decay curves show that the MKLP2-MD neck-linker becomes more ordered in the presence of ADP. AlFx - as shown by the increase in rotational correlation time - compared to the absence of nucleotide, and depends on MT binding (*Figure 6E*).

Strikingly, however, density corresponding to C-terminal residues extending from around H508, which would correspond to a docked neck-linker conformation, is not visible in the ADP.AlFx density map (*Figure 6F*). Residues of the MKLP2-MD extended N-terminus prior to S61 are also not visible. Consistent with a non-canonical neck-linker conformation, the neck-linker sequence itself differs markedly from other plus-end kinesins (*Figure 6D*). Most plus-end directed kinesins have a conserved sequence in which two Asn residues make hydrogen bonds against the core, stabilizing neck-linker docking (*Gigant et al., 2013*) (*Figure 6D*, *Figure 6—figure supplement 2*). In MKLP2, these two Asn residues have been substituted for basic His residues, which would be predicted to preclude neck-linker docking. In addition, conserved core residues contacting this region of the neck-linker in plus-end directed kinesins are not fully conserved in MKLP2; for example Kin1-Y77→MKLP2-Q151 (*Figure 6D*). Thus, although subdomain rearrangement results in reorientation of the MKLP2-MD neck-linker towards the MT plus end – as with other plus end kinesins – the docking of the neck linker that is typically seen does not occur in MKLP2-MD.

## MKLP2 loop6 forms a distinct subdomain

Loop6 is the largest MKLP2-specific insertion (99 aa) and density corresponding to it is present in all nucleotide states (*Figure 7*, *Figure 7—figure supplement 1*). The higher resolution of the ADP.AlFx reconstruction facilitates the characterisation of density corresponding to loop6 (*Figure 7A,B*, *Video 3*), which emerges from the side of the motor domain facing the MT plus end. Given its position, connectivity and its apparent movement in the NN/ADP.AlFx transition, it is effectively part of the SwI/II subdomain (*Figure 7C*). Loop6 forms density that spreads across the MT-facing surface of the central β-sheet. This density also connects the MT facing surface of the core β-sheet with β-sheet-5a/b of the tubulin-binding subdomain that connects to the MT surface (*Figure 7B*). However, loop6 itself does not appear to contact the MT surface. Creating a full model of loop6 was not possible, but secondary structure prediction of loop6 converged on the presence of a 4-turn helical region towards its N-terminus (*Figure 7—figure supplement 1E*). Density that corresponds to a helical structure was observed lying close to helix-α3 and we modelled an α-helix within it (*Figure 7A, B*). However, the rest of loop6 has no predicted secondary structure and may be intrinsically

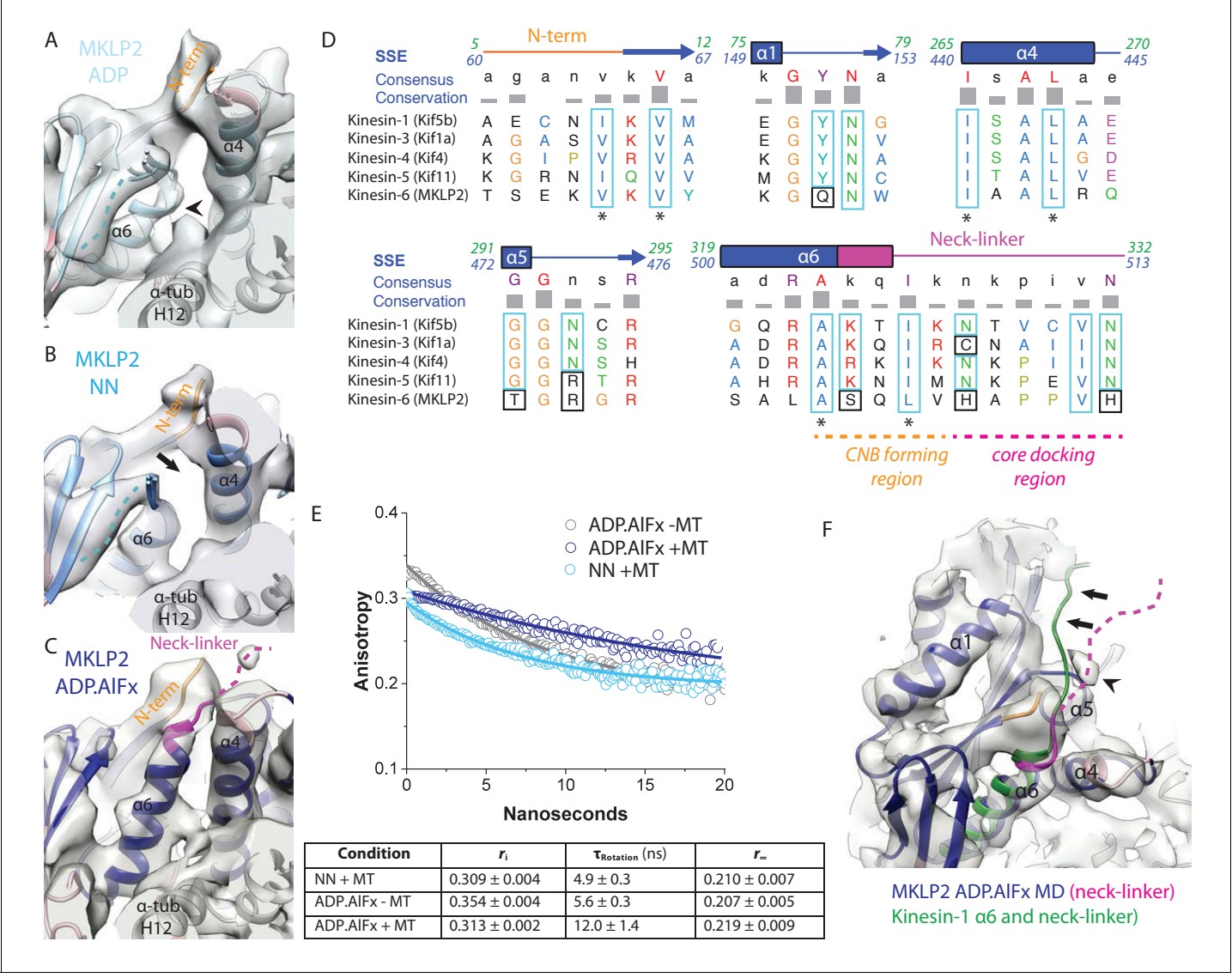

**Figure 6.** MKLP2 neck-linker response to nucleotide. (**A**) MKLP2-MD-ADP has a short helix-α6 leaving a gap between its terminus and helix-α4 (arrow), showing the cleft for the neck linker is closed and the CNB is not formed. Helix-α6 has relatively poor density, suggesting it may be partially disordered at its C-terminus (arrowhead). Density that could flexibly accommodate part of the neck linker is present connected to β-sheet1 (blue dotted line). (**B**) MKLP2-MD-NN also has a short helix-α6 leaving a gap between its terminus and helix-α4 (arrow). (**C**) MKLP2-MD-ADP.AlFx has an extended helix-α6, the initial portion of the neck-linker (magenta) inserts between the N-terminus (orange) and helix-α4 forming the CNB (arrowhead). The remaining portion of the neck-linker is flexible. (**D**) Sequence alignments of representative members (in brackets) of the Kin1/3/4/5/6 families for the neck-linker and its contact regions. Residue colouring using the Clustal X scheme (*Larkin et al., 2007*). Sequence numberings for Kif5b (Kin1, green) and MKLP2 (kinesin-6, blue) are shown adjacent to the secondary structure schematics. Well conserved residues in MKLP2 are boxed in light blue, whereas otherwise well conserved residues which have diverged in MKLP2 are boxed in black (see also *Figure 6—figure supplement 2*). Asterisks indicate conserved hydrophobic residues participating in CNB formation. Regions of the neck-linker involved in CNB formation or core docking are indicated by dashed lines in orange and magenta, respectively. (**E**) Time-resolved fluorescence anisotropy of FlAsH labeled MKLP2-MD: (1) NN + MT, light blue; (2) ADP.AlFx –MT, grey and (3) ADP.AlFx +MT, dark blue. Data shown in the table below are representative of 5 replicate samples. (**F**) MKLP2-MD-ADP. AlFx neck-linker in magenta (disordered region, dashed line). Kin1 helix-α6 and neck-linker (dark green) has been superimposed on helix-α6 of MKLP2-MD, showing the expected position of the neck-linker. There is no density corresponding to a docked neck-linker (arrows) suggesting it is mainly disordered. A small amount of density close to helix-α5 (arrowhead) likely indicates that alternative conformations are flexibly explored.

The online version of this article includes the following figure supplement(s) for figure 6:

**Figure supplement 1.** The neck-linker in MT-bound MKLP2-MD-AMPPNP is not directed towards the MT plus end.

**Figure supplement 2.** MKLP2's neck-linker has an atypical sequence across species that precludes core-docking.

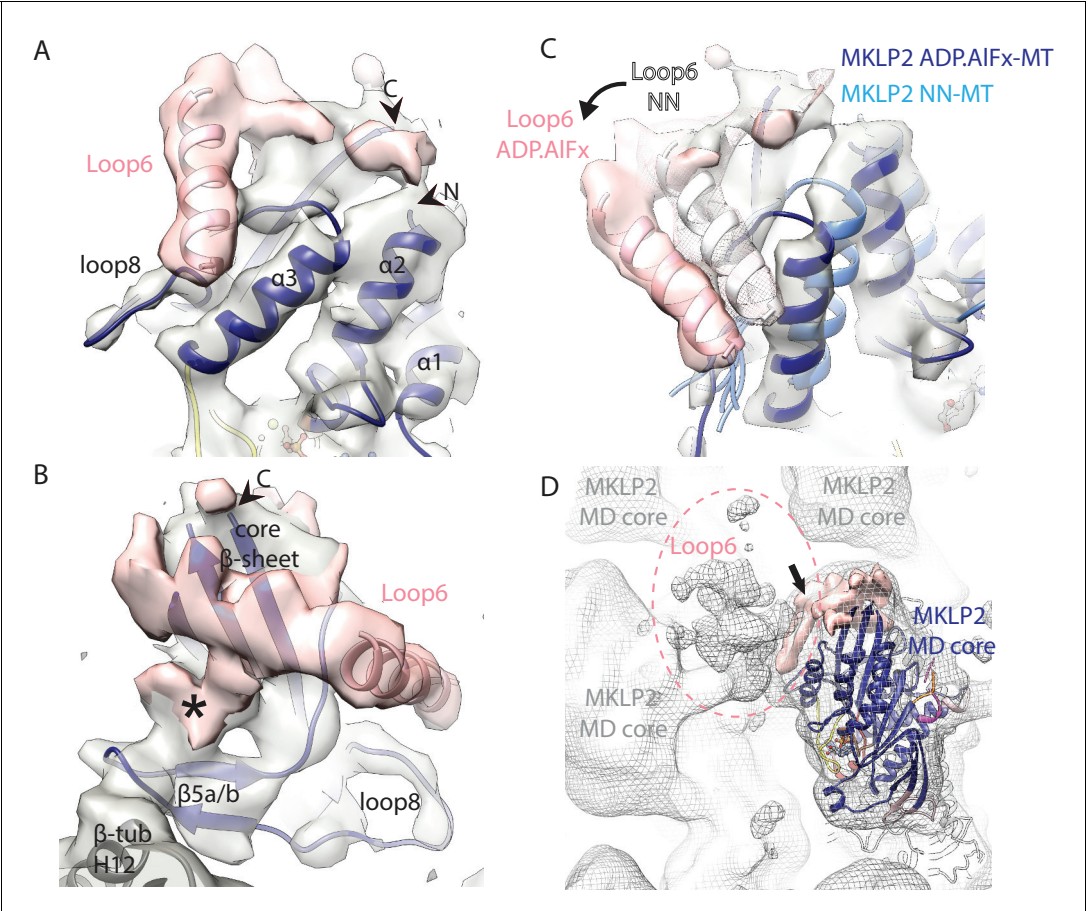

**Figure 7.** Visualisation of MKLP2's loop6 insertion. (**A,B**). MKLP2-MD-ADP.AlFx model (dark blue) within the cryo-EM density (grey) with ordered loop6 density (pink). A model of the predicted α-helix at the loop6 N-terminus fits well into a 'sausage' of loop6 density (pink helix). The termini of loop6 in the model are indicated (arrowheads). (**B**) View from the MT proximal side of the MKLP2-MD. Loop6 density (pink) connects the core β-sheet and β5a/b of the tubulin-binding subdomain (asterisk). (**C**) Movement of loop6 in NN (medium blue) to ADPAlFx (navy blue) transition, with the loop6 helices shown as white or pink ribbons respectively. The MKLP2-MD-ADP.AlFx motor core density is shown as transparent grey solid, MKLP2-MD-ADP.AlFx loop6 density is shown as pink transparent solid and MKLP2-MD-NN loop6 density is shown as pink mesh. (**D**) View of ~4 asymmetric units with the MT plus end towards the top of the panel. Mesh shows density for the MKLP2-MD-ADP.AlFx reconstruction when low pass filtered to ~15 Å. The MKLP2-MD-ADP.AlFx model is docked in one asymmetric unit. After low pass filtering, additional less ordered density attributable to loop6 was observed to the top left of the motor domain (indicated within pink dashed ring). Ordered loop6 density as in the previous panels is shown in pink (and arrowhead). The online version of this article includes the following figure supplement(s) for figure 7:

**Figure supplement 1.** Density visualised for MKLP2's loop6 is not dependent on nucleotide state.

disordered (*Seeger and Rice, 2013*). As displayed in *Figure 7A,B*, loop6 is incompletely visualised, partly due to the lack of discrete secondary structural elements and also presumably due to flexibility in this region of the motor. This conclusion is reinforced when, with coarser (low-pass) filtering of the EM reconstruction, a large and lower resolution cloud of density is revealed (*Figure 7D*), consistent with what was previously described from a 2.5 nm resolution reconstruction of the *C. elegans* kinesin-6, MKLP1 (Zen4) (*Hizlan et al., 2006*). This density is unconnected to other regions of the motor domain, and does not contact adjacent motor domains or the MT, but likely corresponds to multiple flexible conformations of the remaining loop6 residues.

## Discussion

While the cell biological contributions of MKLP2 and other kinesin-6s during mitosis are increasingly well understood, little has been known about the molecular mechanism of these divergent mitotic motors. Using cryo-EM to generate structures that guide comparative modelling and flexible fitting,

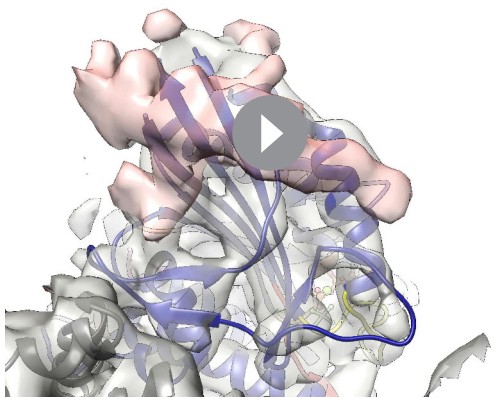

**Video 3.** Loop6 density in the MKLP2-MD-ADP.AlFx cryo-EM reconstruction. Cryo-EM density for the MD core and tubulin are shown in transparent grey solid, whilst additional density attributable to loop6 is shown as transparent pink solid. An α-helix at the N-terminus of loop6, suggested by secondary structure prediction (see Materials and methods and *Figure 7—figure supplement 1E*), which when modelled (pink helix) fits well into a 'sausage' of density attributable to a portion of loop6, is introduced. All density apart from loop6 density is then removed for clarity.

https://elifesciences.org/articles/27793#video3

we have been able to visualise sequential states of the MT-bound MKLP2-MD ATPase cycle. Our structural dissection of a mammalian MKLP2-MD allows identification of aspects of its mechanism that appear conserved among kinesins along with many properties that differ substantially from other members of the superfamily.

Our set of structures allows us to characterise the response of the track-bound MKLP2-MD to nucleotide. Some aspects of its nucleotide-dependent conformational responses appear to be similar to Kin1/3/5, and can be framed in terms of relative movements of subdomains within the motor domain and the clefts that open and close between them (*Video 1*, *Figure 6*, *Figure 3—figure supplement 2*) (*Shang et al., 2014*; *Cao et al., 2014*). The so-called nucleotide cleft (NC) that lies between the Switch I/II and P-loop subdomains is predicted to close in the presence of bound nucleotide and to open in its absence. The polymer cleft (PC) lies between the tubulin-binding and the Switch I/II subdomains; it is closed in the presence of a close interaction between helix-α4 and SwII and open in its absence, and is proposed to change at different points in the MT-bound kinesin ATPase cycle. The docking cleft (DC) lies between the P-loop and tubulin-binding subdomains and when open allows insertion of helix-α6 and reorientation of the neck-linker towards the MT plus end (*Shang et al., 2014*). Thus, for MKLP2-MD: (i) in the ADP-bound structure, the PC is open, while the NC is closed and the DC for the neck-linker is also closed; (ii) in the NN state, the PC is closed while the NC is open, such that the DC remains closed; (iii) in the ADP.AlFx state in which the active site is closed in a catalytic-competent conformation, both the PC and NC are closed leading to an opening of the DC and reorientation of the neck-linker towards the MT plus end. Our MKLP2-MD structures thus reinforce the logic of cleft opening/closure with respect to kinesin mechanochemistry first described for Kin1 (*Shang et al., 2014*), while demonstrating an elaboration of motor function through family-specific modifications.

The biochemistry of MKLP2 is highly divergent compared to canonical kinesins. The $k_{cat}$ of MKLP2-MD (4.4 s$^{-1}$) is ~10 fold lower than kinesin-1 and -3 (*Table 1*) and a number of MKLP2-MD substitutions at otherwise highly conserved regions could influence multiple steps in its catalytic cycle. For example, MKLP2-MD has low ADP affinity in solution and this is barely altered on interaction with MTs. Indeed, the ADP release rate in the absence of MTs is only 2–3-fold less than that for Kin1/3 s in the presence of saturating MTs. Thus, while MT binding activates the MKLP2 ATPase, its basal ADP release rate is higher than $k_{cat}$. The small structural changes observed when comparing our ADP and NN structures – in particular, that the N-terminus of helix-α4 and loop11 remain largely disordered in the NN state – is a distinctive property of these MKLP2/MT complexes compared to other plus end kinesins (*Figure 3—figure supplement 1* [*Atherton et al., 2014*; *Shang et al., 2014*; *Goulet et al., 2014*; *Cao et al., 2014*]). Substitutions at conserved sites around the nucleotide pocket might be predicted to destabilise the nucleotide binding network that is present in other kinesins. For example, substitution in loop9 at a residue otherwise completely conserved in all kinesins (Kin1-R190→MKLP2-Q365, *Figure 3—figure supplement 1A*) could perturb side chain interactions proposed to be part of the so-called 'Mg$^{2+}$' stabiliser' (*Chang et al., 2013*). Indeed, mutagenesis of this network in Kin1 (*Cao et al., 2014*) or Kin3 (*Nitta et al., 2008*) increases ADP release rates in the absence of MTs. Similarly, substitutions in and around the P-loop (Kin1-A83

→MKLP2-T157 and Kin1-Q86 →MKLP2-V160), along with helix-α6 (Kin1-N308→MKLP2-T489) may perturb the stability of loop11/P-loop at the active site through alteration of the so-called 'anchor' network (*Chang et al., 2013*). Loop11 itself is one residue shorter than in other N-kinesins, adopts a different conformation compared to Kin1 and contains substitutions in residues that contribute to stabilisation of Kin1's ordered conformation in the NN and ATP-like states (Kin1-T241→MKLP2-Q417, and Kin1-A243→MKLP2-S419, *Figure 3—figure supplement 1A*). We conclude that the collective effect of these substitutions – along with possible effects from the shift of MKLP2-MD binding on the MT surface – is likely to be the reason for lower stabilisation of nucleotide binding to MKLP2-MD.

Given these properties, it is thus not clear what state our ADP-bound reconstruction corresponds to in terms of intermediates in the MKLP2 ATPase cycle. It could for example be the MKLP2-MD-ADP complex as it binds the MT from solution, or could represent a MT-bound NN structure captured after ADP binding. Given the fast kinetics of ADP release by MKLP2-MD, the latter seems more likely. Therefore, previously proposed mechanisms of MT-stimulated ADP release in Kin1/3 s may not be applicable to MKLP2-MD (*Atherton et al., 2014*; *Shang et al., 2014*; *Cao et al., 2014*). Nevertheless, the position of the P-loop is shifted away from the MT in the MKLP2-NN structure compared to the ADP-bound structure and the PC is closed, conformational changes that were described to coincide with ADP release in Kin1 (*Shang et al., 2014*; *Cao et al., 2014*). While apparently redundant in actively driving ADP release in MKLP2, these cleft and subdomain transitions when ADP is absent may still be important for subsequent larger scale rearrangements in the motor domain on ATP-binding and its conversion towards a hydrolysis-competent state.

The structure of the MKLP2-MD-ADP.AlFx reconstruction suggests that MT binding participates in stabilising helix-α4 and loop11 in this structure, supporting the conserved SwitchI/II and P-loop residues in adopting a catalytically relevant conformation. The MKLP2-ADP.AlFx reconstruction is overall comparable to hydrolysis-competent conformations seen in Kin1/3/4/5 with a so-called phosphate tube formed from loop9 and 11 supporting ATP hydrolysis. However, the reconstruction in the presence of AMPPNP most closely resembles that of ADP – rather than ADP.AlFx as is seen in Kin1/3 – and it may also represent the MKLP2-MD NN conformation after AMPPNP has bound to it. This too could be a reflection of MKLP2-specific substitutions within and around the nucleotide binding site (*Figure 3—figure supplement 1*) that influence the binding kinetics and structural transitions of MT-bound MKLP2-MD in response to different nucleotides, and which also contribute to the overall turnover rate of the motor's ATPase. Precedents for different responses to different nucleotide analogues have also been observed in other kinesins – for example, small differences in motor conformation between AMPPNP and ADP.AlFx were observed for human Kin5 (*Goulet et al., 2014*). Future detailed kinetics will yield further insights into the defining transitions of MKLP2 during its ATPase cycle.

The MKLP2-MD steady state ATPase activity shows that its $K_{0.5,MT}$ is ~1.1 μM. This agrees well with the co-sedimentation analysis, which also shows that, while the affinity is higher in the NN and AMPPNP states, there is less effect of nucleotide on the overall affinity of the MKLP2-MD for MTs compared to Kin1. Our reconstructions do not provide obvious structural explanations for these subtle nucleotide-dependent differences in the MT affinity of MKLP2-MD. Indeed, the additional MKLP2-MD loop contacts with the MT were modelled most easily in the highest resolution reconstruction (ADP.AlFx), which has a slightly lower MT affinity. It is thus possible that the flexible loops formed by the MKLP2-MD specific insertions do not directly contribute to differences in affinity in this motor compared to Kin1; rather this is dominated by the differences in the surface charge of the MT interface (*Figure 5*). Further insight concerning the effects on the kinetics of structural transitions induced by different nucleotides will be required to fully dissect these effects.

MKLP2-MD $K_{0.5,MT}$ is ~10 fold lower than the Kin1 motor domain (13 μM) (*Atherton et al., 2014*). While both motors bind MTs via their conserved tubulin-binding subdomains, strikingly the MKLP2-MD MT interface is overall more positively charged than that of Kin1, which likely contributes to the overall increased affinity of MKLP2 throughout the ATPase cycle. In addition, the MKLP2-specific, nucleotide invariant extension of loop12 is positively charged and may interact with the nearby acidic C-terminal tails (CTTs) of β-tubulin, although there is no visible connection in our reconstructions, likely due to its conformational variability. A similar interaction between the β-tubulin CTT and loop12 of Kin3 has previously been described (*Okada and Hirokawa, 2000*), although Kin3 loop12 has a higher net positive charge (+4) than MKLP2 and is longer and conformationally flexible in all

nucleotide states (*Atherton et al., 2014*). Sequence insertions within loop2 are more usually associated with depolymerisation activity, for example in Kin13s (*Ogawa et al., 2004*), but here MKLP2-MD loop2 appears to simply form a MT contact, with no current evidence of MKLP2 activity influencing MT dynamics. In addition to these loop insertions creating additional MT contacts with the primary tubulin binding site, MKLP2-MD loop8 can reach to contact an adjacent protofilament.

The collective effects of all the differences between MKLP2-MD and Kin1/3 manifest in the striking observation of bound MKLP2-MD shifted rotationally and translationally relative to the MT surface in all nucleotide states, creating a different footprint on the MT surface. This produces altered interactions across the MT surface including in loop11 adjacent to the active site and, together with the MKLP2-specific modifications around the active site, may have a knock-on effect on the catalytic activity of the motor. Thus, while not sufficient to perturb the fundamentals of nucleotide-sensitive subdomain rearrangement or formation of a hydrolysis competent conformation at the active site, MKLP2-MD altered position on the MT surface may regulate the transition rates between structural states. The altered footprint may also enable MKLP2 to select different populations of MT tracks within the spindle for interaction. Thus, our structural characterisation of MKLP2-MD suggests that the binding site of kinesins on the MT surface has some previously unsuspected plasticity that enables tuning of motor function while still supporting conserved conformational changes.

In Kin1/3/5, the key conformational change elicited by ATP binding to MT-associated motors is ordering of the neck-linker towards the MT plus end (*Rice et al., 1999*). Such neck-linker docking is supported by formation of the CNB between the neck-linker and the motor domain N-terminus (*Khalil et al., 2008*). In MKLP2, reorientation of the neck-linker and CNB formation occur in the presence of ADP.AlFx and the neck-linker is less mobile than in other nucleotide states (*Figure 6E*). This is despite the shortness of helix-α6 in the ADP/NN states and substitution of Kin1-K323→MKLP2-S504 at the transition between helix-α6 and neck linker. CNB formation is, however, consistent with conservation across all N-kinesins including MKLP2 of a cluster of hydrophobic residues in this region of the motor domain (*Figure 6* [*Hwang et al., 2008*]). Formation of the CNB also contributes to stable closure of the nucleotide binding pocket and supports ATPase activity (*Cao et al., 2014*).

Remarkably, however, the distal region of the putative MKLP2-MD neck-linker does not visibly dock along the plus end portion of the motor domain core, and demonstrates residual flexibility despite CNB formation. This observation is consistent with the divergent MKLP2 sequences both in the neck-linker and along the core β-sheet where the neck-linker might otherwise dock (*Figure 6D*, *Figure 6—figure supplement 2*) [*Vale and Fletterick, 1997*]). Furthermore, the MKLP2-MD neck-linker also contains two adjacent proline residues (P510 and P511) that are likely to affect the conformation of the neck-linker. Divergence in the distal neck-linker region is a feature of the MKLP2 sequence in a number of organisms (*Figure 6—figure supplement 2*). Indeed, our structural findings suggest that analysis of sequences immediately C-terminal of kinesin motor domains could be considered in two segments according to (1) CNB formation by the proximal region and (2) docking of the distal region, when predicting motor function. It is not known if dimers of MKLP2 operate by taking steps in the same way as characterised for transport kinesins, and our structures predict that the molecular properties of full length MKLP2 are likely to be distinct from Kin1. In addition, the MKLP2 neck region – separating the neck-linker and the coiled coil – is substantially longer (~40 residues) than for example Kin1, making it unlikely that conventional models of processive stepping apply to MKLP2.

An extended loop6 insertion is characteristic of kinesin-6s and was visible in our MKLP2 reconstructions, with its location and its partial conformation visualised both at lower and higher resolutions, respectively (*Figure 7*, *Video 3*). In addition, sequence analysis suggests that the N-terminal helical portion of loop6 is also likely to be conserved in MKLP2s in metazoa (*Figure 7—figure supplement 1E*). During preparation of our manuscript, a structural snapshot of the *C. elegans* kinesin-6 Zen4 motor domain in the absence of its MT/tubulin track was published (*Guan et al., 2017*). Intriguingly, although the loop6 insertion of this MKLP1 relative is substantially shorter than in MKLP2, a helical segment is also observed in this region of Zen4, consistent with our predictions. However, the connectivity of loop6 in the two kinesin-6s is different, with MKLP2 loop6 forming much more extensive contacts with other regions within the SwI/II subdomain. This leaves open the question as to this loop's functional contribution to MKLP2. Given its connectivity with more conserved portions of the motor domain such as loop8/beta-5, it could have an overall modulatory role on motor mechanochemistry. It is also possible that it is the binding site for other regulatory components of the

central spindle. Sequence comparisons informed by our structural data suggest that the presence of loop6, other extended insertions including loop2, and modifications related to neck-linker docking are highly characteristic of this subfamily of motors. Thus, our structural data shed light on the unusual mechanochemistry of the MKLP2-MD and could be used to predict the properties of other members of the kinesin-6 subfamily, including the two other kinesin-6 family members within vertebrates. Similarly, sequence analysis in the structural context of one highly divergent kinesin subfamily can also be mapped onto other members of the wider superfamily.

Our structural and biophysical characterisations of the MKLP2 motor domain suggest that this protein and its relatives could act as an organiser or a tension sensor, rather than a classical transport motor, within the MT bundles at the spindle midzone. A non-transport role for kinesin-6s was also implied by recent computational analysis of the kinesin superfamily (*Richard et al., 2016*). Nevertheless, MKLP2 inhibitor studies highlight the importance of the ATPase activity of MKLP2 for its role within the spindle, suggesting that the nucleotide-dependent conformational changes we have described are important for its mitotic function (*Labrière et al., 2016*). The atypical mechanochemistry of MKLP2 could also facilitate the motor's response to signalling cues throughout cell division, especially as mitosis proceeds towards cytokinesis. In combination with the capacity of its C-terminal region also to bind MTs (*Echard et al., 1998*), teams of MKLP2 motors in the spindle midzone could contribute to collective sliding and organisation of the dense collection of MTs found there. The altered footprint of MKLP2-MD binding on the MT surface may influence MT bundle organisation in this context. The activity of MKLP2 is also subject to numerous points of post-translational modification and regulation by mitotic kinases, including within or near the motor domain. For example, a Cdk1 site (T197) is located within loop6 in the connecting region between the body of the motor domain and the α-helix visualised in our reconstruction (*Kitagawa et al., 2014*), while a Plk1 site/ binding site is in the neck region beyond the neck linker (S527) where other binding partners are known to interact. Such modifications could also fine tune the mechanochemistry of MKLP2 and regulate its precise function. Our characterisation of MT-bound mammalian MKLP2 by cryo-EM sets the structural stage for future studies to further dissect the mechanism, function and regulation of this divergent mitotic kinesin.

## Materials and methods

### Protein expression, purification, and labelling

The MKLP2 constructs were cloned into a pET28 vector with C-terminal 6His tag and expressed in BL21 Gold (DE3) *E. coli* cells by induction with 0.2 mM IPTG at 20°C overnight. Cells were harvested by centrifugation at 4500 $\times$ g for 10 min and lysed by sonication in lysis buffer (50 mM HEPES, pH 7; 500 mM NaCl, 40 mM Imidazole, 1 mM TCEP, 0.5 mM PMSF, 0.1 mM ADP, 5 mM MgCl$_2$). The lysate was clarified at 20,000 $\times$ g for 45 min at 4°C; and the protein was purified by HisTrap column (GE Healthcare Life Science, Pittsburgh, PA). Further purification was achieved by size exclusion chromatography in buffer containing 20 mM HEPES pH 7, 100 mM NaCl, 1 mM TCEP, 5 mM MgCl$_2$, and 0.1 mM ADP. Fractions containing purified MKLP2 protein were pooled and concentrated to ~10 mg/ml, flash-frozen in liquid nitrogen and stored in −80°C until usage. For anisotropy decay studies, the 25–520 MKLP2-MD construct was modified by inserting the sequence CCPGCC at its C-terminus before the His tag sequence. The construct was expressed and purified as above. It was labelled with the fluorescein derivative FlAsH (Molecular Probes, Eugene, OR) by incubation of a 10-fold molar excess of label over protein in 25 mM HEPES, 50 mM KAc, 5 mM MgAc, pH 7.5 for 12 hr followed by removal of unbound label over prepoured Sephadex G25 columns (GE Healthcare). The stoichiometry of labelling was typically 0.8–0.9 FlAsH:MKLP2-MD.

### Steady state ATPase, transient kinetic methodologies

MKLP2 construct ATPase activities were determined, in duplicate, in a buffer containing 50 mM KAc, 25 mM HEPES, 5 mM MgAc, and 1 mM EGTA, pH 7.5 (RT) by measuring released phosphate using a commercial kit (EnzChek Phosphate Assay, Molecular Probes). Binding of the fluorescent nucleotide analogues 2'dmT (in duplicate) and 2'dmD (once) to 4:1 complexes of MTs:MD were measured by mixing with an excess of fluorescent nucleotide in the stopped flow at 20°C. Samples were rendered nucleotide free prior to mixing by incubating for 20 min with 0.2 U/ml apyrase (Type VII,

Sigma Aldrich, St Louis, MO). Fluorescence enhancement of the mant fluorophor was monitored by energy transfer from vicinal tryptophans by exciting at 295 nm and monitoring 90° from the incident beam through a 450 nm broad bandpass filter (Omega Optical, Brattleboro, VT). Data were subjected to linear least squares fitting. Dissociation of 2'dmD from MKLP2-MD was measured by adding a 10-fold molar excess of 2'dmD to MKLP2-MD in ATPase buffer and mixing with 2 mM MgADP. The resulting fluorescence decrease was monitored by energy transfer from vicinal tryptophans by exciting at 295 nm and monitoring 90° from the incident beam through a 450 nm broad bandpass filter (Omega Optical).

## MT co-sedimentation assay

Tubulin (Cytoskeleton Inc, Denver, CO) was resuspended in BRB80 (80 mM PIPES, 2 mM $MgCl_2$, 1 mM EGTA, 1 mM DTT) to 50 µM. Polymerization of tubulin into MTs was carried out by addition of 1 mM GTP and incubated at 37°C for 1 hr, followed by addition of 200 µM paclitaxel (Cytoskeleton) and additional incubation for 1 hr at 37°C. Pacliltaxel-stabilized MTs were kept at room temperature for >24 hr, centrifuged at 15,000 × g for 30 min, and resuspended to 100 µM with BRB80 buffer plus 200 µM paclitaxel. Equilibrium binding experiments were performed at 25°C in BRB80 buffer plus 25 mM NaCl (all concentrations reported as final values after mixing). MKLP2-MD was incubated with ADP or ADP.AlFx for 30 min. For NN sample, MKLP2-MD was first treated with apyrase to remove all bound nucleotides. 2.5 µM MKLP2-MD in different nucleotide state was incubated with paclitaxel-stabilized MTs (0–10 µM polymerized tubulin; 20 µM paclitaxel) in a 100 µl reaction with 1 mM corresponding nucleotide. The mixture was incubated for 30 min and centrifuged at 100,000 × g for 30 min at 25°C. The supernatant was removed and the pellet was gently rinse with warm BRB80 buffer plus 20 µM paclitaxel. The pellet was resuspended in cold BRB80 buffer plus 10 mM $CaCl_2$. The supernatant and pellet from each sample were analysed on SDS-PAGE gels. The concentrations of MKLP2-MD in the supernatant and pellet were quantified using Image J software. GraphPad Prism was used for curve fitting using a quadratic function that assumes 1:1 MD:tubulin binding stoichiometry:

$$MD_{bound} = \frac{(MT_{total} + MD_{total} + K_D) - \sqrt{(MT_{total} + MD_{total} + K_D)^2 - 4(MT_{total})(MD_{total})}}{2}$$

where $MD_{bound}$ is [MKLP2-MD] bound to MTs, $MD_{total}$ is total MKLP2-MD in the assay, $MT_{total}$ is total MT content in the assay, and $K_D$ is the dissociation constant for binding.

## Time-resolved fluorescence anisotropy

Time-resolved fluorescence anisotropy (TFA) data were acquired as described previously (*Muretta et al., 2013*) using 9 µM FlAsH labeled MKLP2-MD (85% labeled) in the presence or absence of 15 µM MT. Single exponential functions to the nanosecond FlAsH anisotropy decays of labelled MKLP2-MD were fitted for: (i) bound to MTs in the absence of nucleotide, (ii) in the presence of ADP.AlFx in solution or (iii) bound to MTs in the presence of ADP.AlFx. The buffer used was 25 mM HEPES, 50 mM KAc, 5 mM MgAc, pH 7.5. Anisotropy decays were acquired from two independent preparations of protein. For each biochemical condition assayed, a minimum of 5 independent samples were acquired. Data were subjected to non-linear optimization to determine best fitting single exponential decay parameters. Data reported represent the mean of parameters from fits of replicate data ± the standard error for the fit parameter.

## Cryo-EM sample preparation

MTs for MKLP2-MD ADP.AlFx, AMPPNP and no nucleotide data sets were polymerized by incubating 50 µM bovine tubulin (Cytoskeleton Inc, Denver, CO) in MES polymerization buffer (100 mM MES, pH 6.5, 1 mM $MgCl_2$, 1 mM EGTA, 1 mM DTT, 5 mM GTP) for 1 hr at 37°C. GMPCPP MTs used for the MKLP2-MD ADP dataset were double-cycled by polymerizing 20 µM bovine tubulin in BRB80 buffer with 1 mM GMPCPP for 45 mins at 37°C, depolymerizing on ice and repolymerizing for a further 45 mins with an additional 2 mM GMPCPP. 1 mM paclitaxel was then added to all MT preps, incubated for a further 1 hr at 37°C, and left at room temperature for 24–48 hr before use. MTs were diluted to a final concentration of 2.5 µM in BRB80 buffer. MKLP2-MD was buffer exchanged into BRB20 (20 mM PIPES, 2 mM $MgCl_2$, 1 mM EGTA, 1 mM DTT) containing either 2

mM of AMPPNP, ADP, ADP +AlF4, or apyrase (10 units/ml), diluted to a final concentration of 60 µM MKLP2-MD and left for 15 min at room temperature before use. MT and MKLP2-MD samples were added in a sequential fashion as 4 µl droplets to glow-discharged C-flat™ holey carbon EM grids (Protochips, Morrisville, NC) before blotting and plunge-freezing in liquid ethane using a Vitrobot (FEI Co., Hillsboro, OR).

## Cryo-EM data collection and data processing

Images of MKLP2-MD ADP, ADP.AlFx and NN states were collected on a FEI Tecnai G2 Polara operating in low dose mode at 300 kV, using a DE20 direct electron detector (Direct Electron, San Diego, CA) with a final sampling of 1.53 Å/pixel. A total electron dose of ~50e-/$\text{Å}^2$ over a 1.5 s exposure at 15 frames/s gave a total of 22 frames, at ~2.2e-/$\text{Å}^2$ per frame. Images of the MKLP2-MD-AMPPNP state were collected on a FEI Tecnai F20 operating in low dose mode at 200 kV using a DE20 direct electron detector (Direct Electron) with a final sampling of 1.54 Å/pixel. A total electron dose of ~40e-/$\text{Å}^2$ over a 1 s exposure at 25 frames/s gave a total of 25 frames, at ~1.6e-/$\text{Å}^2$ per frame.

To correct for sample drift and local beam induced movement respectively, individual frames were globally aligned using IMOD (RRID:SCR_003297) scripts, then locally aligned using the Optical Flow approach implemented in Xmipp (*de la Rosa-Trevín et al., 2013*). The full dose was used for particle picking and CTF determination using CTFFIND3 (*Mindell and Grigorieff, 2003*), while 25e-/$\text{Å}^2$ or 20e-/$\text{Å}^2$ were used for particle alignment, angular assignment and 3D reconstruction on data collected at 300 kV or 200 kV respectively. MT segments selected in Eman *boxer* served as input to a set of custom-designed semi-automated single-particle processing scripts utilizing Spider and Frealign as described previously, with minor modifications (*Atherton et al., 2014*; *Shang et al., 2014*; *Sindelar and Downing, 2010*). Poorly aligned particles were excluded from final reconstructions according to Frealign's reported phase-residual values. The final dataset size (in asymmetric units) are: MKLP2-MD-ADP – 137,788; MKLP2-MD-NN – 120,341; MKLP2-MD-ADP.AlFx – 276,111; MKLP2-MD-AMPPNP – 141,154. The resolutions of symmetrized reconstructions are shown in *Figure 2—figure supplement 1A–D*. The binding surfaces of kinesin/MTs in these final structures were coloured according to the electrostatic potential as calculated with pdb2pqr (*Dolinsky et al., 2007*) and APBS (*Baker et al., 2001*).

## Sequence analysis and modelling of MKLP2-MD

As shown in *Figure 2—figure supplements 3*, 100 homology models for each nucleotide state were generated with MODELLER v9.15 (RRID:SCR_008395) (*Sali and Blundell, 1993*) using multiple templates (*Table 2*). Templates were selected based on sequence identity and structural identity (inferred from secondary structure predictions made using Psipred (RRID:SCR_010246), Jpred, and RaptorX, [*Drozdetskiy et al., 2015*; *McGuffin et al., 2000*; *Källberg et al., 2012*]). In addition, the two crystal structures of kinesin-1-α/βtubulin complexes in ADP.AlFx (*Gigant et al., 2013*) or no nucleotide states (*Cao et al., 2014*) were selected as templates to capture the nucleotide/tubulin-dependent conformations of the motor domain; and were verified as a suitable templates by rigid fitting into MKLP2-MD densities. Given the low sequence identity of MKLP2-MD to the identified templates (28–34%), sequence alignment was initially performed in MUSCLE (RRID:SCR_011812) (*Edgar, 2004*), then manually adjusted, based on conserved motor domain residues identified in the Pfam database (version 29.0) (*Bateman et al., 2004*). For each nucleotide state, the 10 best models from MODELLER were selected using SOAP scoring (*Dong et al., 2013*), and QMEAN (*Benkert et al., 2009*) was then used to determine the top model. Global QMEAN scores indicated the models were of good quality (*Table 2*).

## Flexible fitting of models in cryo-EM reconstructions

The fitted MKLP2-MD models were first refined without tubulin (tubulin density was removed using Segger (*Pintilie et al., 2010*) in Chimera (RRID:SCR_004097) (*Pettersen et al., 2004*). Owing to the size and likely disordered nature of the most of loop6, we did not model it. To avoid loop6 density biasing modelling, the difference mapping function in the TEMPy software package (*Joseph et al., 2016*; *Farabella et al., 2015*) was used to remove the density.

The Fit in Map tool (*Goddard et al., 2007*) in Chimera was used to perform a rigid fit of the homology models into EM maps filtered to appropriate resolution (ADP.AlFx = 5.5 Å, no

nucleotide = 7 Å, ADP = 7 Å, AMPPNP = 8 Å). The quality of fit was assessed using SMOC scoring in the TEMPy. This identified several MKLP2-MD loop insertions associated with poor fit. Where loops had clear corresponding density, 100–500 loop conformers were calculated using MODELLER's loop model class (*Fiser and Sali, 2003*), and were ranked and selected based on improved cross-correlation. A hybrid approach to flexible fitting was performed using simulated annealing molecular dynamics with Flex-EM (*Topf et al., 2008*), and normal mode analysis with iMODFIT (*Lopéz-Blanco and Chacón, 2013*) similar to a previous study (*Pandurangan et al., 2014*). Flexible fitting was performed using secondary structure elements (SSEs) as rigid bodies (and loop regions treated as 'all-atoms'), and the local cross-correlation of SSEs were calculated using the SCCC scoring function in TEMPy. The SCCC scores for Flex-EM and iMODFIT models were compared. To capture the results of both fitting methods, the best fitting SSEs from iMODFIT were combined with the Flex-EM models. Finally, using Flex-EM, an all atom refinement was performed on loops that connected iMODFIT and Flex-EM SSEs. Global cross-correlation analysis showed an increase in score after fitting, and QMEAN analysis showed no degradation in model quality (*Table 2*). In addition, SMOC scores show an increase in local cross-correlation (*Figure 2—figure supplement 4*). For the ADP, NN, and AMPPNP states, there remained several loops with low local cross-correlation scores (7–20% below the mean score). These loops had clear corresponding density, but which was also ambiguous. To represent this, 500–1000 loop conformers were calculated as above. Following clustering based on cross-correlation and Cα RMSD, five conformers from the top cluster were selected.

Finally, tubulin (PDB ID: 1JFF [*Löwe et al., 2001*]) was rigidly fit into the EM densities. The interaction interface between MKLP2-MD and tubulin was calculated via docking driven by HADDOCK (*Dominguez et al., 2003*; *van Zundert et al., 2016*). 50 non-interfacial MKLP2-tubulin residue pairs were selected at random, then the Cα-Cα distances were calculated and used as unambiguous distance restraints to preserve the MKLP2-tubulin rigid fit. Phi/psi angles were also restrained. In addition, density corresponding to loop6 was calculated by generating synthetic density of each MKLP-MD motor core model (i.e. modelled elements apart from loop6) and subtracting it from the relevant experimental density.

## Acknowledgements

JA, CAM (MR/J000973/1), AC (MR/J003867/1), A-PJ and MT (MR/M019292/1) are supported by the Medical Research Council, UK. JMM was supported by funding from the American Heart Association (SDG20480032). SRR is supported by funding from NIH (GM102875 and NS073610). AH was supported by grants from the CNRS, ANR-15-CE13-0017-01, Instruct-pilote, Ligue Contre le Cancer (RS14/75-28), ARC (PJA 20151203285) and INCa (2016–1-PL BIO-10-ICR-1). I-MY has been awarded a Marie Curie Fellowship FP7-PEOPLE-2012-IIF. The AH team is part of Labex CelTisPhyBio 11-LBX-0038, which is part of the Initiative d'Excellence at PSL Research University (ANR-10-IDEX-0001–02 PSL). The authors thank Charles Sindelar (Yale University, USA) for reconstruction algorithms.

## Additional information

### Funding

| Funder | Grant reference number | Author |
| --- | --- | --- |
| European Commission | Marie Curie Fellowship | I-Mei Yu |
| Medical Research Council | MR/J003867/1 | Alexander Cook |
| American Heart Association | SDG20480032 | Joseph M Muretta |
| Medical Research Council | MR/M019292/1 | Maya Topf |
| National Institute of General Medical Sciences | GM102875 | Steven S Rosenfeld |
| National Institute of General Medical Sciences | NS073610 | Steven S Rosenfeld |
| Centre National de la Recherche Scientifique | | Anne Houdusse |

| | | |
|---|---|---|
| Agence Nationale de la Recherche | ANR-15-CE13-0017-01 | Anne Houdusse |
| Ligue Contre le Cancer | RS14/75-28 | Anne Houdusse |
| Instruct-pilote | | Anne Houdusse |
| Australian Research Council | PJA 20151203285 | Anne Houdusse |
| Institut National Du Cancer | 2016–1-PL BIO-10-ICR-1 | Anne Houdusse |
| Medical Research Council | MR/J000973/1 | Carolyn A Moores |
| Marie Curie Fellowship | FP7-PEOPLE-2012-IIF | I-Mei Yu |

The funders had no role in study design, data collection and interpretation, or the decision to submit the work for publication.

## Author contributions

Joseph Atherton, Conceptualization, Formal analysis, Validation, Investigation, Visualization, Methodology, Writing—original draft, Writing—review and editing; I-Mei Yu, Conceptualization, Formal analysis, Investigation, Visualization, Methodology, Writing—original draft, Writing—review and editing; Alexander Cook, Investigation, Visualization, Methodology, Writing—original draft; Joseph M Muretta, Formal analysis, Investigation, Visualization, Writing—original draft; Agnel Joseph, Software, Supervision, Methodology, Writing—original draft; Jennifer Major, Investigation; Yannick Sourigues, Resources, Investigation; Jeffrey Clause, Investigation, Methodology; Maya Topf, Software, Supervision, Funding acquisition, Validation, Investigation, Methodology, Writing—original draft; Steven S Rosenfeld, Anne Houdusse, Conceptualization, Resources, Formal analysis, Supervision, Funding acquisition, Validation, Investigation, Methodology, Writing—original draft, Project administration, Writing—review and editing; Carolyn A Moores, Conceptualization, Supervision, Funding acquisition, Writing—original draft, Project administration, Writing—review and editing

## Author ORCIDs

Carolyn A Moores (iD) http://orcid.org/0000-0001-5686-6290

## Decision letter and Author response

Decision letter https://doi.org/10.7554/eLife.27793.sa1
Author response https://doi.org/10.7554/eLife.27793.sa2

# Additional files

## Supplementary files

• Transparent reporting form

## Data availability

The following datasets were generated:

| Author(s) | Year | Dataset title | Dataset URL | Database and Identifier |
|---|---|---|---|---|
| Atherton J, Yu IM, Cook A, Muretta JM, Joseph AP, Major J, Sourigues Y, Clause J, Topf M, Rosenfeld SS, Houdusse A, Moores CA | 2017 | 14-protofilament microtubule-bound mammalian MKLP2 motor domain with ADP | http://www.ebi.ac.uk/pdbe/entry/emdb/EMD-3620 | Protein Data Bank, EMD-3620 |
| Atherton J, Yu IM, Cook A, Muretta JM, Joseph AP, Major J, Sourigues Y, Clause J, Topf M, Rosenfeld SS, | 2017 | 13-protofilament microtubule-bound mammalian MKLP2 motor domain in the absence of nucleotide | http://www.ebi.ac.uk/pdbe/entry/emdb/EMD-3621 | Protein Data Bank, EMD-3621 |

| | | | | | |
|---|---|---|---|---|---|
| Houdusse A, Moores CA | | | | | |
| Atherton J, Yu IM, Cook A, Muretta JM, Joseph AP, Major J, Sourigues Y, Clause J, Topf M, Rosenfeld SS, Houdusse A, Moores CA | 2017 | 13-protofilament microtubule-bound mammalian MKLP2 motor domain with ADP.AlFx | http://www.ebi.ac.uk/pdbe/entry/emdb/EMD-3622 | Protein Data Bank, EMD-3622 |
| Atherton J, Yu IM, Cook A, Muretta JM, Joseph AP, Major J, Sourigues Y, Clause J, Topf M, Rosenfeld SS, Houdusse A, Moores CA | 2017 | 13-protofilament microtubule-bound mammalian MKLP2 motor domain with AMPPNP | http://www.ebi.ac.uk/pdbe/entry/emdb/EMD-3623 | Protein Data Bank, EMD-3623 |
| Atherton J, Yu IM, Cook A, Muretta JM, Joseph AP, Major J, Sourigues Y, Clause J, Topf M, Rosenfeld SS, Houdusse A, Moores CA | 2017 | 14-protofilament microtubule-bound mammalian MKLP2 motor domain with ADP | http://www.ebi.ac.uk/pdbe/entry/pdb/5nd2 | Protein Data Bank, 5ND2 |
| Atherton J, Yu IM, Cook A, Muretta JM, Joseph AP, Major J, Sourigues Y, Clause J, Topf M, Rosenfeld SS, Houdusse A, Moores CA | 2017 | 13-protofilament microtubule-bound mammalian MKLP2 motor domain in the absence of nucleotide | http://www.ebi.ac.uk/pdbe/entry/pdb/5nd3 | Protein Data Bank, 5ND3 |
| Atherton J, Yu IM, Cook A, Muretta JM, Joseph AP, Major J, Sourigues Y, Clause J, Topf M, Rosenfeld SS | 2017 | 13-protofilament microtubule-bound mammalian MKLP2 motor domain with ADP.AlFx | http://www.ebi.ac.uk/pdbe/entry/pdb/5nd4 | Protein Data Bank, 5ND4 |
| Atherton J, Yu IM, Cook A, Muretta JM, Joseph AP | 2017 | 13-protofilament microtubule-bound mammalian MKLP2 motor domain with AMPPNP | http://www.ebi.ac.uk/pdbe/entry/pdb/5nd7 | Protein Data Bank, 5ND7 |

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
