## [Decision Letter]

Thank you for submitting your article "The divergent mitotic kinesin MKLP2 exhibits atypical structure and mechanochemistry" for consideration by *eLife*. Your article has been reviewed by two peer reviewers, and the evaluation has been overseen by a Reviewing Editor and Michael Marletta as the Senior Editor. The following individuals involved in review of your submission have agreed to reveal their identity: William O Hancock (Reviewer #2); Ken Downing (Reviewer #3).

The reviewers have discussed the reviews with one another and the Reviewing Editor has drafted this decision to help you prepare a revised submission.

Summary:

This paper deals with the results of cryo-EM structural studies of MKLP2 bound to microtubules (MTs), complemented by biophysical measurements that document its unusual properties. The authors present structures of 4 states of the nucleotide hydrolysis cycle, which allow development of several novel insights about the relation of this type of kinesin, kinesin-6, to other, better studied variants. With the 4 reconstructions and a substantial number of insertions in the kinesin-6 sequence, the paper is uncommonly rich in results and discussion of specific parts of the structure. The resolution, in the range of 6 Å for the kinesin, is sufficient to discern much of the secondary structure. However, it still leaves much to be desired in terms of giving confidence to understanding loop regions, as well as several parts of the molecule that retain substantial flexibility even while docked to the MT. Nonetheless, the authors have been able to derive atomic models for most of the protein in each of the states. They have used a series of steps that give reasonable confidence in these models, which are used to gain insight on the mechanics of the nucleotide cycle. While this interpretation in some places seems rather bold, the authors are in fact straightforward about the limitations of their models and interpretations. The models are about as good as can be generated from the current data and overall, the interpretation is quite well founded. The EM results support identification of quite a number of differences in the MKLP2 structure, its changes through the nucleotide cycle, and its interactions with MTs.

Essential revisions:

1) The ADP binding kinetics in Figure 1 (and ATP kinetics in Figure 1) are very interesting, but very difficult to reconcile with more traditional kinesins (as the authors state). The finding that the ADP off-rate is actually faster in the absence of microtubules runs opposite to the kinesin 1/3/5 paradigm. However, because it requires an extrapolation to zero nucleotide, binding of mant ADP to apo motor doesn't seem like the ideal experiment to nail this point. There are a number or predictions of this fast and microtubule-independent ADP binding that can be tested. For instance, in the absence of microtubules, mixing increasing concentrations of mant ADP with motors and measuring the steady state fluorescence should give a predictable curve with a high K_d_ (estimated to be around 50 μm from Figure 1). Alternatively, flushing motors incubated with mant ADP against mM unlabeled ADP should give a more direct measurement of k_off_ for ADP. These ADP binding kinetics are part of the uniqueness of this motor, and it is worth nailing these kinetics down more firmly.

2) For the motor binding data in Figure 1, the use of the quadratic form of the binding isotherm is reasonable, but there is concern about the relatively high motor concentration of 2.5 μm used in the experiments. When the K_d_ is <10-fold smaller than the motor concentration, it is very difficult to discern between different K_d_ values because the curves all look similar to a ramp up to the motor concentration and then a plateau. Hence, the NN and AMPPNP K_d_, which are reported to be 0.04 μM, should be considered upper limits only. Accordingly, points about a smaller difference in affinities between strong and weak binding states as compared to kinesin 1 should be dropped. The most important point in Figure 1 is really the relatively high affinity in the ADP state, and this point still stands because if affinity were significantly weaker than this it would be obvious in the plots. Because there are FLASH labeled motors, using fluorescence rather than gels in microtubule pelleting would allow much lower motor concentrations to be used, allowing a better estimate of K_d_ for NN and AMPPNP.

3) The contrast between the 1 μm KmMt in the ATPase and the tighter K_d_ from the pelleting experiments is puzzling. The ADP and ADP-AlFx K_d_ are about 3-fold lower than the Km from the ATPase, so one answer is that the cycle is dominated by the ADP or ADPPi states and the difference is just from experimental uncertainties. However, the fast ADP off-rate in Figure 1 argues against this model of the hydrolysis cycle – with the reported nucleotide binding kinetics, the motor should be in NN or ATP states most of the time it would seem, which would predict a much lower KmMt from the ATPase experiments.

---

## [Author Response]

Essential revisions:1) The ADP binding kinetics in Figure 1 (and ATP kinetics in Figure 1) are very interesting, but very difficult to reconcile with more traditional kinesins (as the authors state). The finding that the ADP off-rate is actually faster in the absence of microtubules runs opposite to the kinesin 1/3/5 paradigm. However, because it requires an extrapolation to zero nucleotide, binding of mant ADP to apo motor doesn't seem like the ideal experiment to nail this point. There are a number or predictions of this fast and microtubule-independent ADP binding that can be tested. For instance, in the absence of microtubules, mixing increasing concentrations of mant ADP with motors and measuring the steady state fluorescence should give a predictable curve with a high K_d_ (estimated to be around 50 μm from Figure 1). Alternatively, flushing motors incubated with mant ADP against mM unlabeled ADP should give a more direct measurement of k_off_ for ADP. These ADP binding kinetics are part of the uniqueness of this motor, and it is worth nailing these kinetics down more firmly.

As the reviewers have appreciated, the kinetic properties of MKLP2 which emerge from our data highlight this motor as highly unusual. The first experimental approach suggested for additional characterization of the ADP off-rate of this motor, however, is not a practical solution to getting an accurate binding constant for mant ADP: even when we excite the mant fluorophore through FRET by exciting at 292 nm, there is some direct excitation of the mant. Since mant ADP for these experiments needs to be in large molar excess over the MKLP2 motor domain, background fluorescence contribution from the unbound mant nucleotide would be a major limitation. We do, however, wish to thank the reviewers for suggesting the more direct measure of mant ADP dissociation kinetics. As per our revised manuscript, we note (in the Results, Discussion and Materials and methods sections) that mant ADP dissociation in the absence of MTs—at 51.9 ± 0.1 s^-1^ is three orders of magnitude faster than that for other kinesins. These data are shown in a new figure, Figure 1—figure supplement 1. These data indeed round out our kinetic study of this unusual kinesin more firmly.

2) For the motor binding data in Figure 1, the use of the quadratic form of the binding isotherm is reasonable, but there is concern about the relatively high motor concentration of 2.5 μm used in the experiments. When the K_d_ is <10-fold smaller than the motor concentration, it is very difficult to discern between different K_d_ values because the curves all look similar to a ramp up to the motor concentration and then a plateau. Hence, the NN and AMPPNP K_d_, which are reported to be 0.04 μM, should be considered upper limits only. Accordingly, points about a smaller difference in affinities between strong and weak binding states as compared to kinesin 1 should be dropped. The most important point in Figure 1 is really the relatively high affinity in the ADP state, and this point still stands because if affinity were significantly weaker than this it would be obvious in the plots. Because there are FLASH labeled motors, using fluorescence rather than gels in microtubule pelleting would allow much lower motor concentrations to be used, allowing a better estimate of K_d_ for NN and AMPPNP.

We appreciate the reviewers’ comments about these data and agree with them about the potential limitations of data derived from pelleting experiments. It would be a considerable amount of work beyond the scope of the current study to undertake the experiments described with the FLASH labeled motors. We have however altered the text on p6 to reflect the reviewers’ comments about these data.

3) The contrast between the 1 μm KmMt in the ATPase and the tighter K_d_ from the pelleting experiments is puzzling. The ADP and ADP-AlFx K_d_ are about 3-fold lower than the Km from the ATPase, so one answer is that the cycle is dominated by the ADP or ADPPi states and the difference is just from experimental uncertainties. However, the fast ADP off-rate in Figure 1 argues against this model of the hydrolysis cycle – with the reported nucleotide binding kinetics, the motor should be in NN or ATP states most of the time it would seem, which would predict a much lower KmMt from the ATPase experiments.

The 2 experiments that the reviewers seek to compare – steady state ATPase activity and binding constants calculated using co-sedimentation assays – were performed in different buffers which, together with other differences in assay conditions (eg motor:MT ratio) could account for the differences mentioned. The dissection of the origin of the differences of K_d_ vs Km in these disparate experimental setups would be a substantial effort beyond the scope of the current work. The purpose of the currently included data is to a) in the case of the steady ATPase assay establish the comparative activity of different MKLP2 constructs, thereby establishing the validity the structural characterization of the 25-520 construct and b) in the case of the co-sedimentation assay, compare the ADP and ATP-like states. Both data sets thereby support the conclusion that MKLP2 is a highly unusual motor and are thus appropriate for the current study.